# COUNTERFACTUAL FAIRNESS PREDICTION: CONSISTENT ESTIMATION WITH GENERATIVE MODELS AND THEORETICAL GUARANTEES

## ABSTRACT

Fairness in predictions is of direct importance in practice due to legal, ethical, and societal reasons. This is often accomplished through counterfactual fairness, which ensures that the prediction for an individual is the same as that in a counterfactual world under a different sensitive attribute. However, achieving counterfactual fairness is challenging as counterfactuals are unobservable. Existing baselines for counterfactual fairness do not have theoretical guarantees. In this paper, we propose a novel counterfactual fairness predictor for making predictions under counterfactual fairness. Here, we follow the standard counterfactual fairness setting and directly learn the counterfactual distribution of the descendants of the sensitive attribute via tailored neural networks, which we then use to enforce fair predictions through a novel counterfactual mediator regularization. Unique to our work is that we provide theoretical guarantees that our method is effective in ensuring the notion of counterfactual fairness. We further compare the performance across various datasets, where our method achieves state-of-the-art performance.

## 1 INTRODUCTION

Fairness in machine learning is mandated for a large number of practical applications due to legal, ethical, and societal reasons (Barocas & Selbst, 2016; Kleinberg et al., 2019; Feuerriegel et al., 2020; Angwin et al., 2016; De Arteaga et al., 2022; von Zahn et al., 2022). Examples are predictions in credit lending or recidivism prediction, where fairness is mandated by anti-discrimination laws.

In this paper, we focus on the notion of ***counterfactual fairness*** (Kusner et al., 2017). The notion of counterfactual fairness has recently received significant attention (e.g., Kusner et al., 2017; Garg et al., 2019; Xu et al., 2019; Chiappa, 2019; Kim et al., 2021; Abroshan et al., 2022; Garg et al., 2019; Zuo et al., 2022; Grari et al., 2023; Rosenblatt & Witter, 2023; Ma et al., 2023; Zuo et al., 2023; Anthis & Veitch, 2024; Silva, 2024). One reason is that counterfactual fairness directly relates to legal terminology in that a prediction is fair towards an individual if the prediction does not change had the individual belonged to a different demographic group defined by some sensitive attribute (e.g., gender, race). However, ensuring counterfactual fairness is challenging as, in practice, counterfactuals are generally unobservable.

Prior works have developed methods for achieving counterfactual fairness in predictive tasks (see Sec. 2). Originally, Kusner et al. (2017) described a conceptual algorithm to achieve counterfactual fairness. Therein, the idea is to first estimate a set of latent (background) variables and then train a prediction model without using the sensitive attribute or its descendants. More recently, many works have extended the conceptual algorithm through neural methods (Pfohl et al., 2019; Kim et al., 2021; Grari et al., 2023). However, these methods have a key *shortcoming*: they have **no** theoretical guarantees for achieving counterfactual fairness. Our proposed method is the first to ensure consistent estimation of the counterfactual distribution by leveraging the identifiability of counterfactual theory, and further offers theoretical guarantees in achieving counterfactual fairness.

In this paper, we present a novel deep neural network called *Generative Counterfactual Fairness Network* (GCFN) for making predictions under counterfactual fairness. For this, we build upon *Standard Fairness Model* (Plecko & Bareinboim, 2022). Our method leverages tailored generative adversarial networks to directly learn the counterfactual distribution of the descendants of the

sensitive attribute. We then use the generated counterfactuals to enforce fair predictions through a novel *counterfactual mediator regularization*. We further provide theoretical guarantees that, if the counterfactual distribution is learned sufficiently well, our method is effective in ensuring the notion of counterfactual fairness.

Overall, our **main contributions** are as follows:[1] (1) We propose a novel deep neural network for achieving counterfactual fairness in predictions. (2) We further provide theoretical results that our method is guaranteed to ensure counterfactual fairness. (3) We demonstrate that our GCFN achieves the state-of-the-art performance. We further provide a real-world case study of recidivism prediction to show that our method gives meaningful predictions in practice.

## 2 RELATED WORK

### 2.1 FAIRNESS NOTIONS FOR PREDICTIONS

Over the past years, the machine learning community has developed an extensive series of fairness notions for predictive tasks so that one can train unbiased machine learning models; see Appendix B for a detailed overview. In this paper, we focus on *counterfactual fairness* (Kusner et al., 2017), due to its relevance in practice (Barocas & Selbst, 2016; De Arteaga et al., 2022).

### 2.2 PREDICTIONS UNDER COUNTERFACTUAL FAIRNESS

Originally, Kusner et al. (2017) introduced a conceptual algorithm to achieve predictions under counterfactual fairness. The idea is to first infer a set of latent background variables and subsequently train a prediction model using these inferred latent variables and non-descendants of sensitive attributes. Kusner et al. (2017) provided only a conceptual algorithm, while later works offered actual instantiations. The conceptual algorithm can not directly achieve fairness prediction from data and a causal graph; instead, it requires the specification of structural equations to ensure identifiability, which makes it impractical. We provide detailed comparison to Kusner et al. (2017) in Appendix D.1.

State-of-the-art approaches build upon the above idea but integrate neural learning techniques, typically by using VAEs. These are mCEVAE (Pfohl et al., 2019), DCEVAE (Kim et al., 2021), and ADVAE (Grari et al., 2023). In general, these methods proceed by first computing the posterior distribution of the latent variables, given the observational data and a prior on latent variables. Based on that, they compute the implied counterfactual distributions, which can either be utilized directly for predictive purposes or can act as a constraint incorporated within the training loss. In sum, the methods in Pfohl et al. (2019); Kim et al. (2021); Grari et al. (2023) are our main baselines. We provide a detailed comparison of these papers and ours in the Appendix D.2. However, none of these methods have shown the identifiability of the latent variables, which implies non-identifiability of the counterfactual queries, thus can lead to unfair prediction. Further details of the benefits over these latent variable baselines are in Appendix D.3.

**Why existing methods are problematic:** Prior works for counterfactual fairness prediction all act as *heuristics* that may return estimates but these estimates do not correspond to the true value. (e.g., (Pfohl et al., 2019; Kim et al., 2021; Grari et al., 2023; Zuo et al., 2023; Wang et al., 2023; Zhou et al., 2024)). In other words, predictions can be generated that can be unfair. The reason is of theoretical nature: existing baselines did not consider identifiability of counterfactuals and were still unclear under which scenarios counterfactual fairness can be achieved in the first place.

The core of our contribution are solving the following two **shortcomings**: ① The learned latent variables in existing methods are *not* identifiable.[2] The lack of identifiability can lead to that the *true* counterfactual distributions are *not* learned. This can thus lead to an overall *low prediction performance* but, importantly, may even undermine fairness objectives. ②: The existing methods

---

[1]Codes are in the anonymous GitHub: `https://anonymous.4open.science/r/gcfn`. Codes will also be available to a public GitHub repository upon acceptance.

[2]In causal inference, "identifiability" refers to a mathematical condition that permits a causal quantity to be measured from observed data (Pearl, 2009). Importantly, identification is *different* from estimatability because methods that act as heuristics may return estimates but they do not correspond to the true value. For the latter, see D'Amour (2019) where the authors provide several concerns that, if a latent variable is not unique, it is possible to have local minima, which leads to unsafe results in causal inference.

have *no theoretical guarantees whether the method is effective in achieving counterfactual fairness*. We provide further details in Appendix D. To address ①  and ②, *we later offer theoretical guarantees leveraging counterfactual identifiability and that our method is effective in ensuring the notion of counterfactual fairness.*

## 3 PROBLEM SETUP

**Notation:** Capital letters such as $X, A, M$ denote random variables and small letters $x, a, m$ denote their realizations from corresponding domains $\mathcal{X}, \mathcal{A}, \mathcal{M}$. Further, $\mathbb{P}(M)$ is the probability distribution of $M$; $\mathbb{P}(M \mid A = a)$ is a conditional distribution; $\mathbb{P}(M_a)$ the interventional distribution on $M$ when setting $A$ to $a$; and $\mathbb{P}(M_{a'} \mid A = a, M = m)$ the counterfactual distribution of $M$ had $A$ been set to $a'$ given evidence $A = a$ and $M = m$.

We follow the *Standard Fairness Model* (Plecko & Bareinboim, 2022), shown in Fig. 1, where the nodes represent: sensitive attribute $A \in \mathcal{A}$; mediators $M \in \mathcal{M}$ with $\mathcal{M} \subseteq \mathbb{R}$, which are possibly causally influenced by the sensitive attribute; covariates $X \in \mathcal{X}$, which are not causally influenced by the sensitive attribute; and a target $Y \in \mathcal{Y}$, see Appendix. E for more details. We give a practical example of our causal graph in Appendix. E.3. By following the *Standard Fairness Model*, we ensure that our framework matches common applications in recidivism prediction, credit lending, and public resource allocation (Plecko & Bareinboim, 2022).

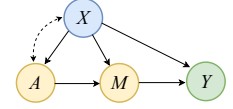

Figure 1: Causal graph. The nodes represent: sensitive attribute $A$; covariate $X$; mediator $M$; target $Y$. $\longrightarrow$ represents a direct causal effect; $\leftarrow\!--\!\rightarrow$ represents the potential presence of hidden confounders.[3]

In our work, $A$ can be a categorical variable with multiple categories $k$ and $X$ and $M$ can be multi-dimensional. For ease of notation, we use $k = 2$, i.e., $\mathcal{A} = \{0, 1\}$, to present our method below. We later present an extension to settings where the sensitive attribute has multiple categories (see Appendix H).

We use the potential outcomes framework (Rubin, 1974) to estimate causal quantities from observational data. Under our causal graph, the dependence of $M$ on $A$ implies that changes in the sensitive attribute $A$ mean also changes in the mediator $M$. We use subscripts such as $M_a$ to denote the potential outcome of $M$ when intervening on $A$. Similarly, $Y_a$ denotes the potential outcome of $Y$. Furthermore, for $k = 2$, $A$ is the factual, and $A'$ is the counterfactual outcome of the sensitive attribute.

Our model follows standard assumptions necessary to identify causal queries (Rubin, 1974). (1) *Consistency:* The observed mediator is identical to the potential mediator given a certain sensitive attribute. Formally, for each unit of observation, $A = a \Rightarrow M = M_a$. (2) *Overlap:* For all $x$ such that $\mathbb{P}(X = x) > 0$, we have $0 < \mathbb{P}(A = a \mid X = x) < 1$, $\forall a \in \mathcal{A}$. (3) *Unconfoundedness:* Conditional on covariates $X$, the potential outcome $M_a$ is independent of sensitive attribute $A$, i.e. $M_a \perp\!\!\!\perp A \mid X$. We discuss the theoretical guarantee on identifiability of counterfactuals under bijective generation mechanisms (BGMs) (Nasr-Esfahany et al., 2023; Melnychuk et al., 2023) in Appendix F.

**Objective:** In this paper, we aim to learn the prediction of a target $Y$ to be *counterfactual fair* with respect to some given sensitive attribute $A$ so that it thus fulfills the notion of *counterfactual fairness* (Kusner et al., 2017). Let $h(X, M) = \hat{Y}$ denote the predicted target from some prediction model, which only depends on covariates and mediators. Formally, our goal is to have $h$ achieve counterfactual fairness if under any context $X = x$, $A = a$, and $M = m$, that is,

$$\mathbb{P}(h(x, M_a) \mid X = x, A = a, M = m) = \mathbb{P}(h(x, M_{a'}) \mid X = x, A = a, M = m). \quad (1)$$

This equation illustrates the need to care about the counterfactual mediator distribution. Under the consistency assumption, the right side of the equality simplifies to the delta (point mass) distribution $\delta(h(x, m))$.

---

[3]The dashed line allows for a correlation between $X$ and $A$ in our framework. Note that, if there is no dashed edge between $X$ and $A$, it is actually a stronger assumption, because it forbids the edge between $X$ and $A$ to have any hidden confounders. However, our setting is more general and allows for the existence of confounders.

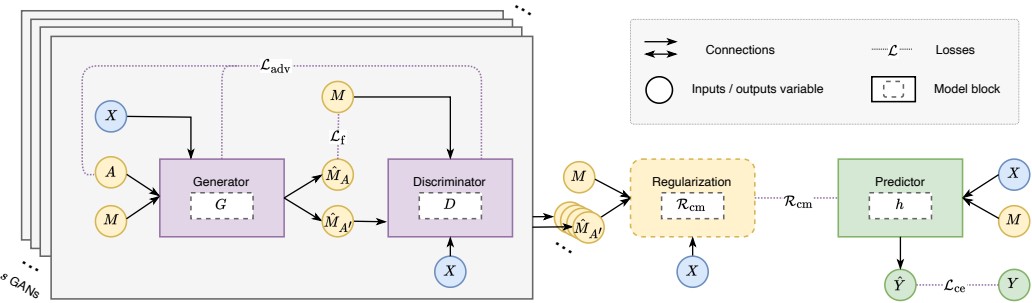

Figure 2: Overview of our GCFN for achieving counterfactual fairness in predictions. *Step 1:* The generator $G$ takes $(X, A, M)$ as input and outputs $\hat{M}_A$ and $\hat{M}_{A'}$. The discriminator $D$ differentiates the observed factual mediator $M$ from the generated counterfactual mediator $\hat{M}_{A'}$. We train $s$ GANs and consider the worst-case counterfactual fairness. *Step 2:* We then use generated counterfactual mediator $\hat{M}_{A'}$ in our counterfactual mediator regularization $\mathcal{R}_{\mathrm{cm}}$. We take the supremum of $\mathcal{R}_{\mathrm{cm}}$ to choose the most 'unfair' generator. Therefore, we enforce the worst-case counterfactual fairness, and the counterfactual mediator regularization $\mathcal{R}_{\mathrm{cm}}$ can enforce the prediction model $h$ to be counterfactual fairness.

## 4 GENERATIVE COUNTERFACTUAL FAIRNESS NETWORK

**Overview:** Here, we introduce our proposed method called *Generative Counterfactual Fairness Network* (GCFN). An overview of our method is in Fig. 2. GCFN proceeds in two steps: **Step 1** uses a significantly modified GAN to learn the counterfactual distribution of the mediator. **Step 2** uses the generated counterfactual mediators from the first step together with our *counterfactual mediator regularization* to enforce counterfactual fairness. The pseudocode is in Appendix G.

*Why do we need counterfactuals of the mediator?* Different from existing methods for causal effect estimation (Yoon et al., 2018; Bica et al., 2020; Zhang et al., 2021), we are *not* interested in obtaining counterfactuals of the target $Y$ ($\hat{=}$ ladder 2 in Pearl's causality ladder). Instead, we are interested in counterfactuals for the mediator $M$, which captures the entire influence of the sensitive attribute and its descendants on the target ($\hat{=}$ ladder 3). Thus, by training the prediction model with our *counterfactual mediator regularization*, we remove the information from the sensitive attribute to ensure fairness while keeping the rest useful information of data to maintain high prediction performance. *What is the advantage of using a GAN in our method?* The GAN in our method enables us to directly learn transformations of factual mediators to counterfactuals without the intermediate step of inferring latent variables. As a result, we eliminate the need for the abduction-action-prediction procedure (Pearl, 2009) and avoid the complexities and potential inaccuracies of inferring and then using latent variables for prediction. Further detailed discussion of the benefits over inferring latent variable baselines in Appendix D.3.

### 4.1 STEP 1: GAN FOR GENERATING COUNTERFACTUAL OF THE MEDIATOR

In Step 1, we aim to generate counterfactuals of the mediator (since the ground-truth counterfactual mediator is unavailable). Our generator $G$ produces the counterfactual of the mediators given observational data. Concurrently, our discriminator $D$ differentiates the factual mediator from the generated counterfactual mediators. This adversarial training process encourages $G$ to learn the counterfactual distribution of the mediator.

#### 4.1.1 COUNTERFACTUAL GENERATOR $G$

The generator $G$ is to learn the counterfactual distribution of the mediator, i.e., $\mathbb{P}(M_{a'} \mid X = x, A = a, M = m)$. Formally, $G : \mathcal{X} \times \mathcal{A} \times \mathcal{M} \rightarrow \mathcal{M}$. $G$ takes the factual sensitive attribute $A$, the factual mediator $M$, and the covariates $X$ as inputs, sampled from the joint (observational) distribution $\mathbb{P}_{X,A,M}$, denoted as $\mathbb{P}_{\mathrm{f}}$ for short. $G$ outputs two potential mediators, $\hat{M}_0$ and $\hat{M}_1$, from which one is factual and the other is counterfactual. For notation, we use $G(X, A, M)$

to refer to the output of the generator. Thus, we have

$$G\left(X, A, M\right)_a = \hat{M}_a \quad \text{for} \quad a \in \{0, 1\}. \tag{2}$$

In our generator $G$, we intentionally output not only the counterfactual mediator but also the factual mediator, even though the latter is observable. The reason is that we can use it to further stabilize the training of the generator. For this, we introduce a reconstructive loss $\mathcal{L}_{\mathrm{f}}$, which we use to ensure that the generated factual mediator $\hat{M}_A$ is similar to the observed factual mediator $M$. Formally, we define the reconstruction loss

$$\mathcal{L}_{\mathrm{f}}(G) = \mathbb{E}_{(X, A, M) \sim \mathbb{P}_{\mathrm{f}}} \left[ \|M - G\left(X, A, M\right)_A\|_2^2 \right], \tag{3}$$

where $\|\cdot\|_2$ is the $L_2$-norm.

### 4.1.2 COUNTERFACTUAL DISCRIMINATOR $D$

The discriminator $D$ is carefully adapted to our setting. In an ideal world, we would have $D$ discriminate between real vs. fake counterfactual mediators; however, the counterfactual mediators are not observable. Instead, we train $D$ to discriminate between factual mediators vs. generated counterfactual mediators. Note that this is different from the conventional discriminators in GANs that seek to discriminate real vs. fake samples (Goodfellow et al., 2014a). Formally, our discriminator $D$ is designed to differentiate the factual mediator $M$ (as observed in the data) from the generated counterfactual mediator $\hat{M}_{A'}$ (as generated by $G$).

We modify the output of $G$ before passing it as input to $D$: We replace the generated factual mediator $\hat{M}_A$ with the observed factual mediator $M$. We denote the new, combined data by $\tilde{G}\left(X, A, M\right)$, which is defined via

$$\tilde{G}\left(X, A, M\right)_a = \begin{cases} M, & \text{if } A = a, \\ G\left(X, A, M\right)_a, & \text{if } A = a', \end{cases} \tag{4}$$

The discriminator $D$ then determines which component of $\tilde{G}$ is the observed factual mediator and thus outputs the corresponding probability. Formally, for the input $(X, \tilde{G})$, the output of the discriminator $D$ is

$$D(X, \tilde{G})_a = \hat{\mathbb{P}}(M = \tilde{G}_a \mid X, \tilde{G}) = \hat{\mathbb{P}}(A = a \mid X, \tilde{G}). \tag{5}$$

### 4.1.3 ADVERSARIAL TRAINING OF OUR GAN

Our GAN is trained in an adversarial manner: (i) the generator $G$ seeks to generate counterfactual mediators in a way that minimizes the probability that the discriminator can differentiate between factual mediators and counterfactual mediators, while (ii) the discriminator $D$ seeks to maximize the probability of correctly identifying the factual mediator. We thus use an adversarial loss $\mathcal{L}_{\mathrm{adv}}$ given by

$$\mathcal{L}_{\mathrm{adv}}(G, D) = \mathbb{E}_{(X, A, M) \sim \mathbb{P}_{\mathrm{f}}} \left[ \log\left( D(X, \tilde{G}\left(X, A, M\right))_A \right) \right]. \tag{6}$$

Overall, our GAN is trained through an adversarial training procedure with a minimax problem as

$$\min_G \max_D \mathcal{L}_{\mathrm{adv}}(G, D) + \alpha \mathcal{L}_{\mathrm{f}}(G), \tag{7}$$

with a hyperparameter $\alpha$ on $\mathcal{L}_{\mathrm{f}}$. Then, under mild identifiability conditions, the counterfactual distribution of the mediator, i.e., $\mathbb{P}\left(M_{a'} \mid X = x, A = a, M = m\right)$, is consistently estimated by our GAN (up to a measure-preserving indeterminacy (Xi & Bloem-Reddy, 2023)), which we state later in Lemma 1.

*Why our method does not simply reproduce the factual mediators?* This is due to the adversarial training. By training the discriminator to differentiate between factual and generated counterfactual mediators, the generator is guided to learn the correct counterfactual distribution. It could be seen as a form of *teacher forcing*. The input of the discriminator $D$ contains the counterfactual and the factual and the order of them is randomized. In our framework, the data are intentionally shuffled, so that factual and counterfactual positions are random. Later, we show this also empirically (Appendix K.2).

## 4.2 STEP 2: COUNTERFACTUAL FAIR PREDICTION THROUGH COUNTERFACTUAL MEDIATOR REGULARIZATION

In Step 2, we use the output of several GANs to train a prediction model under counterfactual fairness in a supervised way. For this, we introduce our *counterfactual mediator regularization* that enforces counterfactual fairness w.r.t the sensitive attribute. Let $h$ denote our prediction model (e.g., a neural network). We define our *counterfactual mediator regularization* $\mathcal{R}_{\mathrm{cm}}(h, G)$ as

$$\mathcal{R}_{\mathrm{cm}}(h, G) = \mathbb{E}_{(X,A,M)\sim\mathbb{P}_{\mathrm{f}}}\left[\left\|h\left(X, M\right) - h\left(X, \hat{M}_{A'}\right)\right\|_2^2\right], \tag{8}$$

where $\hat{M}_{A'} = G(X, A, M)_{A'}$. Our counterfactual mediator regularization has three important characteristics: (1) It is non-trivial. Different from traditional regularization, our $\mathcal{R}_{\mathrm{cm}}$ is not based on the representation of the prediction model $h$ but it *involves a GAN-generated counterfactual $\hat{M}_{A'}$ that is otherwise not observable*. (2) Our $\mathcal{R}_{\mathrm{cm}}$ is not used to constrain the learned representation (e.g., to avoid overfitting) but it is used to change the actual learning objective to achieve the property of counterfactual fairness. (3) Our $\mathcal{R}_{\mathrm{cm}}$ fulfills theoretical properties. Specifically, we show later that, under some conditions, our regularization actually optimizes against counterfactual fairness and thus should learn our task as desired.

The overall loss $\mathcal{L}(h)$ is as follows. We fit the prediction model $h$ using a cross-entropy loss $\mathcal{L}_{\mathrm{ce}}(h)$. We further integrate the above counterfactual mediator regularization $\mathcal{R}_{\mathrm{cm}}(h, G)$ into our overall loss $\mathcal{L}(h)$. For this, we introduce a weight $\lambda \geq 0$ to balance the trade-off between prediction performance and the level of counterfactual fairness. Formally, we have

$$\mathcal{L}(h) = \mathcal{L}_{\mathrm{ce}}(h) + \lambda \sup_{G\in\mathcal{G}} \mathcal{R}_{\mathrm{cm}}(h, G), \tag{9}$$

where $\mathcal{G}$ is a set of all the generators, minimizing Eq. 7, and the supremum over this set chooses the most 'unfair' generator. Therefore, we enforce a *worst-case counterfactual fairness*, as the ground-truth counterfactual distribution is only identifiable up to a measure-preserving indeterminacy (see Appendix F). A large value of $\lambda$ increases the weight of $\mathcal{R}_{\mathrm{cm}}$, thus leading to a prediction model that is strict with regard to counterfactual fairness, while a lower value allows the prediction model to focus more on producing accurate predictions. As such, $\lambda$ offers additional flexibility to decision-makers as they tailor the prediction model based on the fairness needs in practice.

**Considerations of computational efficiency:** Having multiple GANs is primarily to meet the mathematical assumptions and thus ensure theoretical guarantees. We later show that a single GAN is sufficient in practical applications and achieves state-of-the-art performance (see Appendix K.3).

## 4.3 THEORETICAL RESULTS

**Remark 1.** *Let the observational distribution $\mathbb{P}_{X,A,M} = \mathbb{P}_{\mathrm{f}}$ be induced by an SCM $\mathcal{M} = \langle\mathbf{V}, \mathbf{U}, \mathcal{F}, \mathbb{P}(\mathbf{U})\rangle$ with*

$$\begin{aligned}
\mathbf{V} &= \{X, A, M, Y\}, \quad \mathbf{U} = \{U_{XA}, U_M, U_Y\}, \\
\mathcal{F} &= \{f_X(u_{XA}), f_A(x, u_{XA}), f_M(x, a, u_M), f_Y(x, m, u_Y)\}, \\
\mathbb{P}(\mathbf{U}) &= \mathbb{P}(U_{XA})\mathbb{P}(U_M)\mathbb{P}(U_Y),
\end{aligned} \tag{10}$$

*and with the causal graph as in Figure 1. Let $\mathcal{M} \subseteq \mathbb{R}$ and $f_M$ be a bijective generation mechanism (BGM) (Nasr-Esfahany et al., 2023; Melnychuk et al., 2023), i.e., $f_M$ is a strictly increasing (decreasing) continuously-differentiable transformation wrt. $u_M$. Then: The counterfactual distribution of the mediator simplifies to one of two possible point mass distributions*

$$\mathbb{P}(M_{a'} \mid X = x, A = a, M = m) = \delta(\mathbb{F}^{-1}(\pm\mathbb{F}(m; x, a) \mp 0.5 + 0.5; x, a')),$$

*where $\mathbb{F}(\cdot; x, a)$ and $\mathbb{F}^{-1}(\cdot; x, a)$ are a CDF and an inverse CDF of $\mathbb{P}(M \mid X = x, A = a)$, respectively, and $\delta(\cdot)$ is a Dirac-delta distribution. Thus, it is identifiable up to a measure-preserving indeterminacy (Xi & Bloem-Reddy, 2023).*

**Lemma 1** (Consistent estimation of the counterfactual distribution with GAN (up to a measure-preserving indeterminacy))**.** *If the generator of GAN is a continuously differentiable function with respect to $M$, then it consistently estimates one of the counterfactual distributions of the mediator (Eq. 11).*

*Proof.* Intuitively, we first prove that, given an optimal discriminator, the generator of our GAN estimates the distribution of potential mediators for counterfactual sensitive attributes. We then prove that the outputs of the deterministic generator, conditional on the factual mediator $M = m$, estimate $\mathbb{P}(M_{a'} \mid X = x, A = a, M = m)$.) Details are in Appendix F. □

**Remark 2.** *The generator converges to one of the two BGM solutions in Eq. 11. Notably, the difference between the two solutions is negligibly small, when the conditional standard deviation of the mediator is small (see Appendix F).*

Lemma 1 states that our generator consistently estimates the counterfactual distribution of the mediator $\mathbb{P}(M_{a'} \mid X = x, A = a, M = m)$, up to a measure-preserving indeterminacy. The discussion about the assumptions is in Appendix C.2. To the best of our knowledge, we are the first to leverage the identifiability of counterfactual theory to ensure the consistent estimation of the counterfactual distribution by GANs.

Below, we provide theoretical analysis to show that our proposed counterfactual mediator regularization is effective in ensuring counterfactual fairness for predictions. Following Grari et al. (2023), we measure the level of counterfactual fairness $CF$ via $\mathbb{E}\left[\|(h(X, M) - h(X, M_{A'}))\|_2^2\right]$. It is straightforward to see that, the smaller $CF$ is, the more counterfactual fairness the prediction model achieves.

We show that we can quantify to what extent counterfactual fairness $CF$ is fulfilled in the prediction model. We give an upper bound in the following lemma.

**Lemma 2** (Counterfactual mediator regularization bound). *Given the prediction model $h$ that is Lipschitz continuous with a Lipschitz constant $\mathcal{C}$, we have*

$$\mathbb{E}\left[\|(h(X, M) - h(X, M_{A'})\|_2^2\right] \leq \mathcal{C}\,\mathbb{E}\left[\left\|M_{A'} - \hat{M}_{A'}\right\|_2^2\right]$$
$$+ \sup_{G \in \mathcal{G}} \mathcal{R}_{\mathrm{cm}}(h, G), \text{ for every } G \in \mathcal{G}, \tag{11}$$

*where $\hat{M}_{A'} = G(X, A, M)_{A'}$ and $\mathcal{G}$ is a set of all the generators, minimizing the Eq. 7.*

*Proof.* See Appendix F, where we make use of the triangle inequality and several further transformation. □

The inequality in Lemma 2 states that the influence from the sensitive attribute on the target variable is upper-bounded by (i) the estimation of counterfactual mediators (first term) and (ii) the counterfactual mediator regularization (second term). (i) The first term does not depend on $h$ and, given Lemma 1, reduces to zero as there exists a generator in $\mathcal{G}$, which consistently estimates counterfactuals. Hence, by reducing (ii) the second term $\mathcal{R}_{\mathrm{cm}}$ for all the generators through minimizing our training loss in Eq. 9, we can effectively enforce the predictor to learn counterfactual fair predictions.[4]

## 5 EXPERIMENTS

### 5.1 SETUP

**Baselines:** We compare our method against the following state-of-the-art approaches: (1) **CFAN** (Kusner et al., 2017): Kusner et al.'s algorithm with additive noise where only non-descents of sensitive attributes and the estimated latent variables are used for prediction; (2) **CFUA** (Kusner et al., 2017): a variant of the algorithm which does not use the sensitive attribute or any descents of the sensitive attribute; (3) **mCEVAE** (Pfohl et al., 2019): adds a maximum mean discrepancy to regularize the generations in order to remove the information the inferred latent variable from sensitive information; (4) **DCEVAE** (Kim et al., 2021): a VAE-based approach that disentangles the exogenous uncertainty into two variables; (5) **ADVAE** (Grari et al., 2023): adversarial neural learning approach which should be more powerful than penalties from maximum mean discrepancy but is aimed the continuous setting; (6) **HSCIC** (Quinzan et al., 2022): originally designed to enforces the predictions to remain invariant to changes of sensitive attributes using conditional kernel mean

---

[4]Details how we ensure Lipschitz continuity in $h$ are in Appendix J.

embeddings but which we adapted for counterfactual fairness. We also adapt applicable baselines from fair dataset generation: (7) **CFGAN** (Xu et al., 2019): which we extend with a second-stage prediction model. Details are in Appendix J.

**Performance metrics:** Methods for causal fairness aim at both: (i) achieve high accuracy while (ii) ensuring causal fairness, which essentially yields a multi-criteria decision-making problem. To this end, we follow standard procedures and reformulate the multi-criteria decision-making problem using a utility function $U_\gamma(accuracy, CF) : \mathbb{R}^2 \mapsto \mathbb{R}$, where $CF$ is the metric for measuring counterfactual fairness from Sec. 4.2. We define the utility function as $U_\gamma(accuracy, CF) = accuracy - \gamma \times CF$ with a given utility weight $\gamma$. A larger utility $U_\gamma$ is better. The weight $\gamma$ depends on the application and is set by the decision-maker; here, we report results for a wide range of weights $\gamma \in \{0.1, \ldots, 1.0\}$.

**Implementation details:** To ensure our GCFN to achieve counterfactual fairness, we train $s = 10$ GANs and consider the worst-case counterfactual fairness, i.e., we take the maximum value of counterfactual mediator regularization $\max_{j=1}^s \mathcal{R}_{\text{cm}}(h, G_j)$. We implement our GCFN in PyTorch. Both the generator and the discriminator are designed as deep neural networks. We use LeakyReLU, batch normalization in the generator for stability, and train all the GANs for 300 epochs with 256 batch size. The prediction model is a multilayer perceptron, which we train for 30 epochs at a 0.005 learning rate. Since the utility function considers two metrics, the weight $\lambda$ is set to 0.5 to get a good balance. More implementation details and hyperparameter tuning are in Appendix J.

## 5.2 RESULTS FOR (SEMI-)SYNTHETIC DATASETS

We explicitly focus on (semi-)synthetic datasets, which allow us to compute the true counterfactuals and thus validate the effectiveness of our method. We follow previous works that simulate a fully synthetic dataset for performance evaluations (Kim et al., 2021; Quinzan et al., 2022). We find that our GCFN is effective. Detailed results are in Appendix K.1

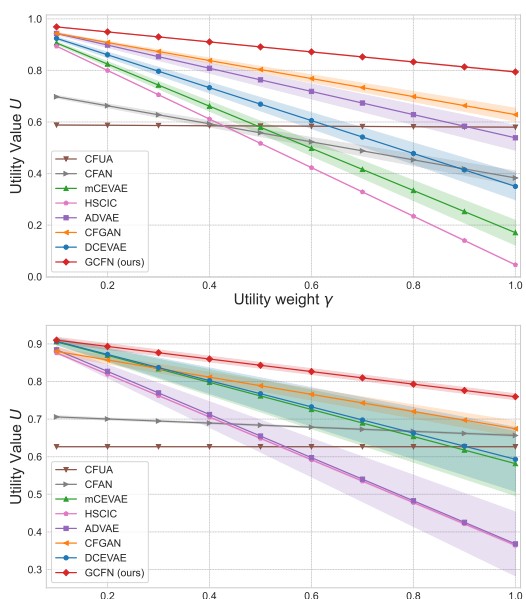

### 5.2.1 RESULTS FOR LSAC DATASET

**Setting:** The Law School (LSAC) dataset (Wightman, 1998) contains information about the law school admission records. We use the LSAC dataset to construct semi-synthetic datasets with two different data-generating mechanisms. (See Appendix I). We predict whether a candidate passes the bar exam and where *gender* is the sensitive attribute. We simulate 101,570 samples and use 20% as the test set.

**Results:** Results are shown in Fig. 3. We make the following findings. (1) Our GCFN performs best. (2) Compared to the baselines, the performance gain from our GCFN is large (up to ~30%). (3) The performance gain for our

Figure 3: Results for LSAC dataset with two different data-generating mechanisms. Utility value $U$: the higher (↑) the better. Shown: mean ± std over 5 runs.

GCFN tends to become larger for larger $\gamma$. (4) Most baselines in the semi-synthetic dataset with sin function have a large variability across runs as compared to our GCFN, which further demonstrates the robustness of our method. (5) Conversely, the strong performance of our GCFN in the semi-synthetic dataset with sin function demonstrates that our tailed GAN can even capture complex counterfactual distributions.

### 5.2.2 ADDITIONAL INSIGHTS

**Computational efficiency:** Our proposed framework used multiple GANs primarily to meet the mathematical assumptions and thus ensure theoretical guarantees. However, having multiple GANs is

not necessary in practice. For this, we provide experimental results in Appendix K.3 to demonstrate that a single GAN is sufficient for state-of-the-art performance.

**Why does our method not simply reproduce the factual mediator?** Intuitively, one may think that the GAN would simply copy the factual mediators because the counterfactual mediators can not be observed during training. However, this is not the case due to the adversarial training process of the generator. By training the discriminator to differentiate between factual and generated counterfactual mediators, the generator is guided to learn the correct counterfactual distribution. It could be seen as a form of teacher forcing.

We conduct experiments to empirically verify that our method does not simply reproduce the factual mediators, but actually learns the counterfactual mediators. Experimental results and details are in Appendix K.2. We observe that the generated counterfactual mediator is similar to the ground-truth counterfactual mediator, while the factual and the generated counterfactual mediators are highly dissimilar. In other words, our model can correctly learn the counterfactual mediators. We additionally give three important arguments for the reason why it works in Appendix K.2.

### 5.3 RESULTS FOR REAL-WORLD DATASETS

We now demonstrate the applicability of our method to real-world data. Since ground-truth counterfactuals are unobservable for real-world data, we refrain from benchmarking, but, instead, we now provide additional insights to offer a better understanding of our method.

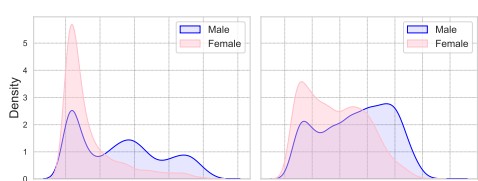

Figure 4: Density of the predicted target variable (salary) across male vs. female. Left: w/o our $\mathcal{R}_{\text{cm}}$. Right: w/ our $\mathcal{R}_{\text{cm}}$.

#### 5.3.1 RESULTS FOR UCI ADULT DATASET

**Setting:** We use UCI Adult (Asuncion & Newman, 2007) to predict if individuals earn a certain salary but where *gender* is a sensitive attribute. Further details are in Appendix I.

**Insights:** To better understand the role of our counterfactual mediator regularization, we trained prediction models both with and without applying $\mathcal{R}_{\text{cm}}$. Our primary focus is to show the shifts in the distribution of the predicted target variable (salary) across the sensitive attribute (gender). The corresponding density plots are in Fig. 4. One would expect the distributions for males and females should be more similar if the prediction is fairer. However, we do not see such a tendency for a prediction model without our counterfactual mediator regularization. In contrast, when our counterfactual mediator regularization is used, both distributions are fairly similar as desired. Further visualizations are in Appendix K.

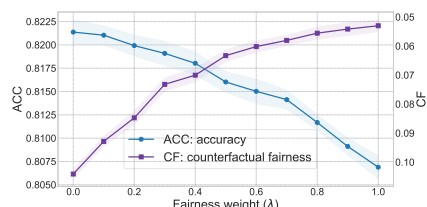

Figure 5: Trade-off between accuracy (ACC) and counterfactual fairness (CF) across different $\lambda$. ACC: the higher ($\uparrow$) the better. CF: the lower ($\downarrow$) the better.

**Accuracy and fairness trade-off:** We vary the fairness weight $\lambda$ from 0 to 1 to see the trade-off between prediction performance and the level of counterfactual fairness. Since the ground-truth counterfactual is not available for the real-world dataset, we use the generated counterfactual to measure counterfactual fairness on the test dataset. The results are in Fig. 5. In line with our expectations, we see that larger values for $\lambda$ lead the predictions to be more strict w.r.t counterfactual fairness, while lower values allow the predictions to have greater accuracy. Hence, the fairness weight $\lambda$ offers flexibility to decision-makers, so that they can tailor our method to the fairness needs in practice.

#### 5.3.2 RESULTS ON COMPAS DATASET

**Setting:** We use the COMPAS dataset (Angwin et al., 2016) to predict recidivism risk of criminals and where *race* is a sensitive attribute. The dataset also has a COMPAS score for that purpose, yet it was revealed to have racial biases (Angwin et al., 2016). In particular, black defendants were

frequently overestimated of their risk of recidivism. Motivated by this finding, we focus our efforts on reducing such racial biases. Further details about the setting are in Appendix I.

**Insights:** We first show how our method adds more fairness to real-world applications. For this, we compare the recidivism predictions from the criminal justice process against the actual reoffenses two years later. Specifically, we compute (i) the accuracy of the official COMPAS score in predicting reoffenses and (ii) the accuracy of our GCFN in predicting the outcomes. The results are in Table 1. We see that our GCFN has a better accuracy. More important is the false positive rate (FPR) for black defendants, which measures how

Table 1: Comparison of predictions against actual reoffenses.

| Method | ACC | PPV | FPR | FNR |
|---|---|---|---|---|
| COMPAS | 0.6644 | 0.6874 | 0.4198 | 0.2689 |
| GCFN (ours) | 0.6753 | 0.7143 | 0.3519 | 0.3032 |

ACC (accuracy); PPV (positive predictive value); FPR (false positive rate); FNR (false negative rate).

often black defendants are assessed at high risk, even though they do not recidivate. Our GCFN reduces the FPR of black defendants from 41.98% to 35.19%. In sum, our method can effectively decrease the bias towards black defendants.

We now provide insights at the defendant level to better understand how black defendants are treated differently by the COMPAS score vs. our GCFN. Fig. 6 shows the number of such different treatments across different characteristics of the defendants. (1) Our GCFN makes oftentimes different predictions for black defendants with a medium COMPAS score around 4 and 5. However, the predictions for black defendants with a very high or low COMPAS score are similar, potentially because these are 'clear-cut'

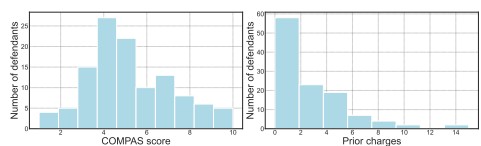

Figure 6: Distribution of black defendants that are treated differently using our GCFN. Left: COMPAS score. Right: Prior charges.

cases. (2) Our method arrives at significantly different predictions for patients with low prior charges. This is expected as the COMPAS score overestimates the risk and is known to be biased (Angwin et al., 2016). Further insights are in the Appendix K.

To exemplify the above, Fig. 7 shows two defendants from the data. Both primarily vary in their race (black vs. white) and their number of prior charges (2 vs. 7). Interestingly, the COMPAS score coincides with race, while our method makes predictions that correspond to the prior charges.

# 6 DISCUSSION

**Limitations & future work:** As with all research on algorithmic fairness, we usher for a cautious, responsible, and ethical use. Sometimes, unfairness may be historically ingrained and require changes beyond the algorithmic layer such as changing sociotechnical parts around data collection or deployment (De Arteaga et al., 2022). The BGM assumption is valid only for real-valued random variables. We thus leave to feature work to prove the counterfactual identifiability in high-dimensionality. **Broader impact:** We offer a novel method for counterfactual

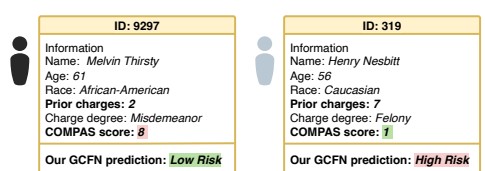

Figure 7: Examples of how defendants are treated differently by the COMPAS score vs. our GCFN.

fairness, which may help to reduce bias against minorities and other marginalized groups. We expect that our theoretical guarantees are especially useful from a regulatory perspective and to promote trust among users.

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

# A MATHEMATICAL BACKGROUND

**Notation:** Capital letters such as $U$ denote a random variable and small letters $u$ its realizations from corresponding domains $\mathcal{U}$. Bold capital letters such as $\mathbf{U} = \{U_1, \ldots, U_n\}$ denote finite sets of random variables. Further, $\mathbb{P}(Y)$ is the distribution of a variable $Y$.

**SCM:** A structural causal model (SCM) (Pearl, 2009) is a 4-tuple $\langle \mathbf{V}, \mathbf{U}, \mathcal{F}, \mathbb{P}(\mathbf{U}) \rangle$, where $\mathbf{U}$ is a set of exogenous (background) variables that are determined by factors outside the model; $\mathbf{V} = \{V_1, \ldots, V_n\}$ is a set of endogenous (observed) variables that are determined by variables in the model (i.e., by the variables in $\mathbf{V} \cup \mathbf{U}$); $\mathcal{F} = \{f_1, \ldots, f_n\}$ is the set of structural functions determining $\mathbf{V}$, $v_i \leftarrow f_i(\text{pa}(v_i), u_i)$, where $\text{pa}(V_i) \subseteq \mathbf{V} \backslash V_i$ and $U_i \subseteq \mathbf{U}$ are the functional arguments of $f_i$; $\mathbb{P}(\mathbf{U})$ is a distribution over the exogenous variables $\mathbf{U}$.

**Potential outcome:** Let $X$ and $Y$ be two random variables in $\mathbf{V}$ and $\mathbf{u} = \{u_1, \ldots, u_n\} \in \mathcal{U}$ be a realization of exogenous variables. The potential outcome $Y_x(\mathbf{u})$ is defined as the solution for $Y$ of the set of equations $\mathcal{F}_x$ evaluated with $\mathbf{U} = \mathbf{u}$ (Pearl, 2009). That is, after $\mathbf{U}$ is fixed, the evaluation is deterministic. $Y_x(\mathbf{u})$ is the value variable $Y$ would take if (possibly contrary to observed facts) $X$ is set to $x$, for a specific realization $\mathbf{u}$. In the rest of the paper, we use $Y_x$ as the short for $Y_x(\mathbf{U})$.

**Observational distribution:** A structural causal model $\mathcal{M} = \langle \mathbf{V}, \mathbf{U}, \mathcal{F}, \mathbb{P}(\mathbf{U}) \rangle$ induces a joint probability distribution $\mathbb{P}(\mathbf{V})$ such that for each $Y \subseteq \mathbf{V}$, $\mathbb{P}^{\mathcal{M}}(Y = y) = \sum_u \mathbb{1}(Y(\mathbf{u}) = y) \mathbb{P}(\mathbf{U} = \mathbf{u})$ where $Y(\mathbf{u})$ is the solution for $Y$ after evaluating $\mathcal{F}$ with $\mathbf{U} = \mathbf{u}$ (Bareinboim et al., 2022).

**Counterfactual distributions:** A structural causal model $\mathcal{M} = \langle \mathbf{V}, \mathbf{U}, \mathcal{F}, \mathbb{P}(\mathbf{U}) \rangle$ induces a family of joint distributions over counterfactual events $Y_x, \ldots, Z_w$ for any $Y, Z, \ldots, X, W \subseteq \mathbf{V}$ : $\mathbb{P}^{\mathcal{M}}(Y_x = y, \ldots, Z_w = z) = \sum_u \mathbb{1}(Y_x(\mathbf{u}) = y, \ldots, Z_w(\mathbf{u}) = z) \mathbb{P}(\mathbf{U} = \mathbf{u})$ (Bareinboim et al., 2022). This equation contains variables with different subscripts, which syntactically represent different potential outcomes or counterfactual worlds.

**Causal graph:** A graph $\mathcal{G}$ is said to be a causal graph of SCM $\mathcal{M}$ if represented as a directed acyclic graph (DAG), where (Pearl, 2009; Bareinboim et al., 2022) each endogenous variable $V_i \in \mathbf{V}$ is a node; there is an edge $V_i \longrightarrow V_j$ if $V_i$ appears as an argument of $f_j \in \mathcal{F}$ ($V_i \in \text{pa}(V_j)$); there is a bidirected edge $V_i \leftarrow\!\!\dashrightarrow V_j$ if the corresponding $U_i, U_j \subset \mathbf{U}$ are correlated ($U_i \cap U_j \neq \varnothing$) or the corresponding functions $f_i, f_j$ share some $U_{ij} \in \mathbf{U}$ as an argument.

## B  FAIRNESS BACKGROUND

### B.1  FAIRNESS NOTIONS

Recent literature has extensively explored different fairness notions (e.g., Dwork et al., 2012; Feldman et al., 2015; Grgic.Hlaca et al., 2016; Hardt et al., 2016; Joseph et al., 2016; Zafar et al., 2017; Wadsworth et al., 2018; Madras et al., 2018; Zhang et al., 2018; Pfohl et al., 2019; Salimi et al., 2019; Celis et al., 2019; Chen et al., 2019; Madras et al., 2019; Di Stefano et al., 2020) For a detailed overview, we refer to Makhlouf et al. (2020) and Plecko & Bareinboim (2022). There have been also theoretical advances (e.g., Fawkes et al., 2022; Rosenblatt & Witter, 2023) but these are orthogonal to ours.

Existing fairness notions can be loosely classified into notions for group- and individual-level fairness, as well as causal notions, some aim at path-specific fairness (e.g., Nabi & Shpitser, 2018; Chiappa, 2019). We adopt the definition of counterfactual fairness from Kusner et al. (2017).

**Counterfactual fairness definition** (Kusner et al., 2017): Given a predictive problem with fairness considerations, where $A, X$ and $Y$ represent the sensitive attributes, remaining attributes, and output of interest respectively, for a causal model $\mathcal{M} = \langle \mathbf{V} = \{A, X, Y\}, \mathbf{U}, \mathcal{F}, \mathbb{P}(\mathbf{U}) \rangle$, prediction model $\hat{Y} = h(X, A, \mathbf{U})$ is counterfactual fair, if under any context $X = x$ and $A = a$,

$$\mathbb{P}\left(\hat{Y}_a(\mathbf{U}) \mid X = x, A = a\right) = \mathbb{P}\left(\hat{Y}_{a'}(\mathbf{U}) \mid X = x, A = a\right), \tag{12}$$

for any value $a'$ attainable by $A$. This is equivalent to the following formulation:

$$\mathbb{P}\left(h(X_a(\mathbf{U}), a, \mathbf{U}) \mid X = x, A = a\right) = \mathbb{P}\left(h(X_{a'}(\mathbf{U}), a', \mathbf{U}) \mid X = x, A = a\right). \tag{13}$$

Our paper adapts the later formulation by doing the following. First, we make the prediction model independent of the sensitive attributes $A$, as they could only make the predictive model unfairer. Second, given the general non-identifiability of the posterior distribution of the exogenous noise, i.e., $\mathbb{P}(\mathbf{U} \mid X = x, A = a)$, we consider only the prediction models dependent on the observed covariates. Third, we split observed covariates $X$ on pre-treatment covariates (confounders) and post-treatment covariates (mediators). Thus, we yield our definition of a fair predictor in Eq. 1.

# C    IDENTIFICATION OF COUNTERFACTUALS

## C.1    IMPORTANCE OF THEORETICAL GUARANTEES FOR IDENTIFAIBILITY

In causal inference, "identifiability" refers to a mathematical condition that permits a causal quantity to be measured from observed data (Pearl, 2009). Importantly, identification is *different* from estimatability because methods that act as heuristics may return estimates but they do not correspond to the true value. For the latter, we refer to D'Amour (2019) where the authors point out several technical issues that, if a latent variable is not unique, it is possible to have local minima, which leads to unsafe results in causal inference. As a result, the lack of identifiability can lead to that the *true* counterfactual distributions is *not* being learned (but some other distribution). This can thus cause an overall *low prediction performance* but, more importantly, may even undermine fairness objectives.

Needless to say, identification of counterfactuals is a very challenging, and current literature on this direction requires assumptions to fulfill it (Xia et al., 2022; Melnychuk et al., 2023; Zhang et al., 2022; Nasr-Esfahany et al., 2023). We are aligned and thus make similar assumptions. Thereby, we can – for the first time – show in which scenarios counterfactual fairness can be fulfilled (and in which scenarios not). This broadens our understanding of how counterfactual fairness operates.

## C.2    DISCUSSION ABOUT BGM ASSUMPTION

The bijective generation mechanism (BGM) (Nasr-Esfahany et al., 2023) is required to ensure the identifiability of our method. The BGM assumption is crucial for the identification of the counterfactuals. It includes many popular identifiable SCMs as special cases, e.g., ANM (Peters et al., 2014), LSNM and (Immer et al., 2023), and PNL (Zhang & Hyvärinen, 2010). Thus, this assumption can be seen as one of the most general assumptions that lead to point identifiability.

It is hard to argue whether real-world datasets usually satisfy the BGM assumption. Rather, this assumption provides a guideline for which datasets it is – in principle – possible to provide answers to the counterfactual questions and for which not. As discovered by (Melnychuk et al., 2023), the relaxation of the BGM assumption not only immediately leads to point non-identifiability but also to non-informative partial identification bounds. Still, it can be intuitively re-formulated (Melnychuk et al., 2023) as follows: In $f_M$, the sensitive attribute, $A$, is assumed to interact only with the observed covariates, X, and not with the exogenous noise, $U_M$. Many real-world data-generation mechanisms / phenomena, if studied closely, can be said to satisfy this assumption (e.g., simulators in physics and medicine but also neuroscience and behavioral processes).

Current literature on counterfactual fairness does not even have any theoretical guarantees on the identifiability of their methods or provide no guarantees of the correctness of counterfactual fairness achieved on either synthetic datasets or real-world datasets. We believe having some assumptions to make our method with theoretical guarantees is better than methods with no correctness guarantee at all and thus helps us to understand where and when counterfactual fairness can be fulfilled from a theoretical point of view.

# D RELATED WORKS

## D.1 KUSNER COUNTERFACTUAL FAIRNESS

Originally, Kusner et al. (2017) introduced a conceptual algorithm to achieve predictions under counterfactual fairness. The idea is to first infer a set of latent background variables and subsequently train a prediction model using these inferred latent variables and non-descendants of sensitive attributes. Kusner et al. (2017) provided only a conceptual algorithm, while only later works proceeded by offering actual instantiations. In particular, Kusner et al. (2017) did not clarify how to learn latent variables in practice and did not prove the identifiability of the inferred latent variables by a model. As such, the conceptual algorithm does not provide any theoretical guarantees for achieving counterfactual fairness in the final prediction models. Furthermore, the conceptual algorithm is not able to directly learn fairness prediction from a given dataset and a given causal graph, but instead it requires the specification of additional structural equations and thus requires further domain knowledge. Without knowing the ground-truth structural causal model, Kusner et al. (2017) can not have identifiability and can not achieve counterfactual fairness. In sum, this makes the conceptual algorithm – and any other instantiation building upon it – impractical.

## D.2 DISCUSSION ABOUT THEORETICAL GUARANTEES IN COUNTERFACTUAL FAIRNESS PREDICTION

Current works related to counterfactual fairness prediction often state that their proposed methods satisfy counterfactual fairness under certain conditions (Pfohl et al., 2019; Kim et al., 2021; Grari et al., 2023; Zuo et al., 2023; Wang et al., 2023; Zhou et al., 2024). However, none of these papers consider or discuss the identifiability of counterfactuals. These papers either require some implied strong assumptions about the identifiability of counterfactuals (which they do not specify) or fully ignore the identifiability at all, and hope their method can somehow manage to learn the correct counterfactuals. As we discuss below, this is often not true because of which these methods may learn predictions that are unfair.

Zuo et al. (2023) claims that they form the counterfactual prediction by drawing from the counterfactual distribution. However, it does not clarify how they exactly managed to learn this counterfactual distribution, and neither does the paper prove why the learned counterfactual distribution by this model should be identifiable. They just give a conceptual algorithm, yet which may learn an unfair objective.

Wang et al. (2023); Zhou et al. (2024) involve the step of learning the latent variables and later training a prediction model using these inferred latent variables. These papers do not clarify how they managed to learn the latent variables and do not have any proper identification guarantees for learned latent variables. For example, they can use, for example, a VAE to estimate the latent variable but it is not guaranteed that it can be correctly identified. Again, this leads to predictions that can be actually unfair.

Methods that act as heuristics as those above may return estimates but these estimates do not correspond to the true value. So, theoretically, these methods can converge against predictions that are not fair but unfair.

## D.3 BENEFITS OVER LATENT VARIABLE BASELINES

Importantly, the latent variable baselines for counterfactual fairness (e.g., mCEVAE (Pfohl et al., 2019), DCEVAE (Kim et al., 2021), and ADVAE (Grari et al., 2023)) are far from being easy as they do not rely on off-the-shelf methods. Rather, they also learn a latent variable in non-trivial ways. The inferred latent variable $U$ should be independent of the sensitive attribute $A$ while representing all other useful information from the observation data. However, there are two main challenges: (1) The latent variable $U$ is *not* identifiable. (2) It is very *hard* to learn such $U$ to satisfy the above independence requirement, especially for high-dimensional or other more complicated settings. Hence, we argue that baselines based on some custom latent variables are highly challenging.

Because of (1) and (2), there are **no** theoretical guarantees for the VAE-based methods. Hence, it is mathematically unclear whether they actually learn the correct counterfactual fair predictions.

In fact, there is even rich empirical evidence that VAE-based methods are often *suboptimal*. VAE-based methods use the estimated variable $U$ in the first step to learn the counterfactual outcome $\mathbb{P}\left(\hat{Y}_{a'}(\mathbf{U}) \mid X = x, A = a, M = m\right)$. The inferred, non-identifiable latent variable can be correlated with the sensitive attribute which may harm fairness, or it might not fully represent the rest of the information from data and harm prediction performance.

Importantly, the latent variable baselines do *not* allow for identifiability. In causal inference, "identifiability" refers to a mathematical condition that permits a causal quantity to be measured from observed data (Pearl, 2009). Importantly, identification is *different* from estimation because methods that act as heuristics may return estimates but they do not correspond to the true value. For the latter, see (D'Amour, 2019) where the authors provide several concerns that, if a latent variable is not unique, it is possible to have local minima, which leads to unsafe results in causal inference.

Non-identifiable for VAE-based methods have been shown in prior works of literature. In a recent paper (Xia et al., 2022), the authors show that VAE-based counterfactual inference do *not* allow for identifiability. The results directly apply to variational inference-based methods, which *do* not have proper identification guarantees. Also, the result from non-linear ICA (which is the task of variational autoencoders) shows that the latent variables are *non-identifiable* (Khemakhem et al., 2020). In simple words, VAE-based methods can estimate the latent variable but it is *not* guaranteed that it can be correctly identified. Note that non-identifiability of the latent variables means non-identifiability of the counterfactual queries. We refer to paper (Melnychuk et al., 2023), which show that the non-identifiability of the latent variables means non-identifiability of the counterfactual queries. Hence, VAE-based methods can **not** ensure that they correctly learn counterfactual fairness, only our method does so.

### D.4 DEEP GENERATIVE MODELS FOR ESTIMATING CAUSAL EFFECTS:

There are many papers that leverage generative adversarial networks and variational autoencoders to estimate causal effects from observational data (Louizos et al., 2017; Kocaoglu et al., 2018; Yoon et al., 2018; Pawlowski et al., 2020; Bica et al., 2020; Zhang et al., 2021; Ma et al., 2024). We borrow some ideas of modeling counterfactuals through deep generative models, yet we emphasize that those methods aim at estimating causal effects but *without* fairness considerations.

### D.5 GENERATING FAIR SYNTHETIC DATASETS

A different literature stream has used generative models to create fair synthetic datasets (e.g., Xu et al., 2018; 2019; van Breugel et al., 2021; Rajabi & Garibay, 2022). Importantly, the task and objective here are *different* from ours. Here, relevant to us is only one method called CFGAN (Xu et al., 2019). However, it is vastly different from our method in many aspects, see difference comparison below.

### D.6 DIFFERENCE FROM CFGAN

Even though CFGAN also employs GANs, it is vastly **different** from our method.

1. *Different tasks:* CFGAN is designed for fair data generation tasks, while our model is designed for learning predictors to be counterfactual fairness. Hence, both address **different tasks**. The training objectives are **different**: CFGAN learns to **mimic factual data**. In our method, the generator **learns the counterfactual distribution of the mediator** through the discriminator distinguishing factual from counterfactual mediators.

2. *Different architectures:* CFGAN employs **two** generators, each aimed at simulating the original causal model and the interventional model, and two discriminators, which ensure data utility and causal fairness. We only employ a streamlined architecture with a **single** generator and discriminator. Further, fairness enters both architectures at **different places**. In CFGAN, fairness is ensured through the GAN setup, whereas our method ensures fairness in a second step through our counterfactual mediator regularization.

3. *Different mathematical objectives:* CFGAN is proposed to synthesize a dataset that satisfies counterfactual fairness in the sampled data. However, a recent paper (Abroshan et al., 2022) has shown that CFGAN is actually considering interventions (=level 2 in Pearl's

causality ladder) and **not** counterfactuals (=level 3).[5] Hence, CFGAN does **not** fulfill the counterfactual fairness notion, but a different notion based on do-operator (intervention). For details, we refer to (Abroshan et al., 2022), Definition 5 therein, called "Discrimination avoiding through causal reasoning"): A generator is said to be fair if the following equation holds: for any context $A = a$ and $X = x$, for all value of $y$ and $a' \in \mathcal{A}$, $P(Y = y \mid X = x, do(A = a)) = P(Y = y \mid X = x, do(A = a'))$, which is different from the counterfactual fairness $P\left(\hat{Y}_a = y \mid X = x, A = a\right) = P\left(\hat{Y}_{a'} = y \mid X = x, A = a\right)$..

4. *No theoretical guarantee for CFGAN:* CFGAN **lacks theoretical support** for its methodology (no identifiable guarantee or counterfactual fairness level). In contrast, our method strictly satisfies the principles of counterfactual fairness and provides theoretical guarantees on the counterfactual fairness level. In sum, **only our method offers theoretical guarantees** for the task at hand.

5. *Suboptimal performance of CFGAN:* Even though CFGAN can, in principle, be applied to counterfactual fairness prediction, it is **suboptimal**. The reason is the following. Unlike CFGAN, which generates complete synthetic data under causal fairness notions, our method only generates counterfactuals of the mediator as an intermediate step, resulting in minimal information loss and better inference performance than CFGAN. Furthermore, since CFGAN needs to train the dual-generator and dual-discriminator together and optimize two adversarial losses, it is more difficult for stable training, and thus its method is less robust than ours.

In sum, even though CFGAN also employs GANs, it is **vastly different from our method**.

---

[5]In the context of Pearl's causal hierarchy**, interventional and counterfactual queries are completely different concepts (Bareinboim et al., 2022). (1) Interventional queries are located on level 2 of Pearl's causality ladder. Interventional queries are of the form $P(y \mid do(x))$. Here, the typical question is "What if? What if I do X?", where the activity is "doing". (2) Counterfactual queries are located on level 3 of Pearl's causality ladder. Counterfactual queries are of the form $P(y_x \mid x', y')$, where $x'$ and $y'$ are different values that $X, Y$ took before. Here, the typical question is "What if I had acted differently?", where the activity is "imagining" had a different treatment selected been made in the beginning. Hence, the main difference is that the counterfactual of $y$ is conditioned on the post-treatment outcome (factual outcome) of $y$ and a different $x$ (where $x$ takes a different value than $x'$). For details, we kindly refer to paper (Bareinboim et al., 2022; Pearl, 2009) for a more technical definition of why intervention and counterfactual are two entirely different concepts.

# E  Causal graph

## E.1  Selection of the causal graph

We choose a causal graph in the way that we do not assume complete knowledge about the causal graph, just whether a variable is pre- or post-treatment, that is whether a variable is a descendant of the sensitive attribute. Furthermore, our assumption of the causal graph is aligned with a standard approach in counterfactual fairness, i.e., with the *Standard Fairness Model* in (Plecko & Bareinboim, 2022)

## E.2  Further clarification

In this section, we clarify why our framework is flexible and broadly applicable.

### E.2.1  Arrow from $X$ to $A$

Below, we clarify that the arrow from $X$ to $A$ is not a limitation but a more general setting.

In causal inference, having an arrow means there is *no* additional assumption, while the absence of an arrow means that there is an assumption. So in our causal graph, the presence of an arrow from $X$ to $A$ indicates the allowance of a causal relationship between variables, that is $X$ can be a direct cause of $A$, this is permissible/allowed in our framework, but the direct causal relationship $X \to A$ is not a necessity. If there is no arrow from $X$ to $A$, it is actually a stronger assumption, because it forbids $X$ to be the cause of $A$.

### E.2.2  Mediator selection

In this section, we clarify that having the mediator in the causal graph is a standard setting (Schröder et al., 2023). Below, we give a detailed introduction to the mediator selection and further offer a step-by-step process on how practitioners can adapt their settings to our graph.

1. Knowledge of $M$ is required for *any* theoretically grounded method that aims to identify/estimate counterfactual effects. It is well-known in the causal inference literature that knowledge of pre-treatment and post-treatment variables is required to identify a causal effect (even in Layer 2 of Pearl's causal hierarchy) (Pearl, 2009). Hence, such knowledge must be available to identify stronger fairness notions such as counterfactual fairness, which lie on layer 3 of Pearl's hierarchy. Works that do not distinguish between post- and pre-treatment variables have no hope of achieving identifiability on the interventional level, and also no hope for the (harder) counterfactual level. The assumption is *consistent* with established literature on causal fairness and the corresponding model is even called "the *Standard Fairness Model*" in the literature (Plecko & Bareinboim, 2022). Mediators are part of a standard causal model, particularly in the context of fairness where sensitive attributes are often involved due to some discrimination-related issues. Mediators naturally occur in practical applications where there are sensitive attributes. Knowledge of $M$ is given in many practical scenarios where our method could be applied.

2. Knowledge of mediators is often realistic as they are usually available mediators in observed data (De Arteaga et al., 2022). The reason is located in the definition of sensitive attributes. Sensitive attributes are of concern to (dis)advantage certain groups of society because the sensitive attribute is known to influence other variables, which may act as proxies that drive some outcome. For example, consider a loan application and take gender as a sensitive attribute. Obviously, gender is sensitive in loan applications as it is known to not only be responsible for discrimination but it naturally affects many other secondary outcomes (e.g., income) which make it so important to mitigate discrimination with respect to the sensitive attribute in the first place. If gender was not affecting outcomes broadly, one would not necessarily see the importance of mitigating discrimination with regard to it and one would thus not consider it as a sensitive attribute.

3. Our assumptions regarding $M$ are weaker than in some existing literature: it is important to clarify that we have a more general setting about mediators. Our method does not require complete knowledge of the entire causal graph. In some previous works, such as (Kim et al.,

2021), detailed knowledge of the causal relationships among mediators is required, while our methodology does not need to specify the causal relationship among them. This makes our framework more flexible and broadly applicable.

4. What if the mediator M is not correctly identified? In this case, one could still employ a "worst-case" approach by including all variables we are unsure about into $M$, which gives us a heuristic worst-case approach, This is similar to other methods that do not distinguish between $X$ and $M$ (Xu et al., 2019). However, those methods that use such heuristics (including baselines) have no theoretical grounding. In summary, we would like to emphasize that the ability of our method to incorporate knowledge of $M$ is an advantage rather than a disadvantage of our method. If $M$ can be correctly identified, then our method provides strong performance (accuracy) with theoretical guarantees on identifiability of counterfactual fairness. If not, the worst-case approach is still possible for applying our method.

In summary, it is important to clarify that we have a more *general setting* about mediators. Our method does not require complete knowledge of the entire causal graph. Our primary objective is to identify whether variables are pre- or post-treatment, in other words, determining if they are descendants of the sensitive attribute. Variables potentially affected by the sensitive attribute are classified as mediators. In some previous works, detailed knowledge of the causal relationships among mediators is required, while our methodology does not. This makes our framework *more flexible and broadly applicable.*

### E.3 PRACTICAL EXAMPLE OF OUR CAUSAL GRAPH

In practice, it is common and typically straightforward to choose which variables act as mediators $M$ through domain knowledge (Nabi & Shpitser, 2018; Kim et al., 2021; Plecko & Bareinboim, 2022). Hence, mediators $M$ are simply all variables that can potentially be influenced by the sensitive attribute. All other variables (except for $A$ and $Y$) are modeled as covariates $X$. For example, consider a job application setting where we want to avoid discrimination by gender. Then, $A$ is gender, and $Y$ is the job offer. Mediators are, for instance, education level or work experience, as both are potentially influenced by gender. In contrast, age is a covariate because it is not influenced by gender.

# F    THEORETICAL RESULTS

## F.1    RESULTS ON COUNTERFACTUAL CONSISTENCY

The natural question arises under which conditions our generator produces consistent counterfactuals. In the following, we provide a theory based on bijective generation mechanisms (BGMs) (Nasr-Esfahany et al., 2023; Melnychuk et al., 2023) and identifiability results for generative models (Xi & Bloem-Reddy, 2023).

**Lemma 3** (Consistent estimation of the counterfactual distribution with GAN (up to a measure-preserving indeterminacy)). *Let the observational distribution $\mathbb{P}_{X,A,M} = \mathbb{P}_{\mathrm{f}}$ be induced by an SCM $\mathcal{M} = \langle \mathbf{V}, \mathbf{U}, \mathcal{F}, \mathbb{P}(\mathbf{U}) \rangle$ with*

$$\mathbf{V} = \{X, A, M, Y\}, \quad \mathbf{U} = \{U_{XA}, U_M, U_Y\},$$
$$\mathcal{F} = \{f_X(u_{XA}), f_A(x, u_{XA}), f_M(x, a, u_M), f_Y(x, m, u_Y)\}, \quad \mathbb{P}(\mathbf{U}) = \mathbb{P}(U_{XA})\mathbb{P}(U_M)\mathbb{P}(U_Y),$$

*and with the causal graph as in Figure 1. Let $\mathcal{M} \subseteq \mathbb{R}$ and $f_M$ be a bijective generation mechanism (BGM) (Nasr-Esfahany et al., 2023; Melnychuk et al., 2023), i.e., $f_M$ is a strictly increasing (decreasing) continuously-differentiable transformation wrt. $u_M$. Then:*

1. *The counterfactual distribution of the mediator simplifies to one of two possible point mass distributions*

$$\mathbb{P}(M_{a'} \mid X = x, A = a, M = m) = \delta(\mathbb{F}^{-1}(\pm\mathbb{F}(m; x, a) \mp 0.5 + 0.5; x, a')), \quad (14)$$

    *where $\mathbb{F}(\cdot; x, a)$ and $\mathbb{F}^{-1}(\cdot; x, a)$ are a CDF and an inverse CDF of $\mathbb{P}(M \mid X = x, A = a)$, respectively, and $\delta(\cdot)$ is a Dirac-delta distribution. Thus, it is identifiable up to a measure-preserving indeterminacy (Xi & Bloem-Reddy, 2023).*

2. *If the generator of GAN is a continuously differentiable function with respect to $M$, then it consistently estimates one of the counterfactual distributions of the mediator (Eq. 14).*

*Proof.* The first statement of the theorem is the main property of bijective generation mechanisms (BGMs), i.e., they allow for deterministic (point mass) counterfactuals. For a more detailed proof, we refer to Lemma B.2 in (Nasr-Esfahany et al., 2023) and to Corollary 3 in (Melnychuk et al., 2023). Importantly, under mild conditions[6], this result holds in the more general class of BGMs with non-monotonous continuously differentiable functions.

The second statement can be proved in two steps. (i) We show that, given an optimal discriminator, the generator of our GAN estimates the distribution of potential mediators for counterfactual sensitive attributes, i.e., $\mathbb{P}(G(x, a, M_a)_{a'} \mid X = x, A = a) = \mathbb{P}(M_{a'} \mid X = x, A = a)$ in distribution. (ii) Then, we demonstrate that the outputs of the deterministic generator, conditional on the factual mediator $M = m$, estimate $\mathbb{P}(M_{a'} \mid X = x, A = a, M = m)$.

(i) Let $\pi_a(x) = \mathbb{P}(A = a \mid X = x)$ denote the propensity score. The discriminator of our GAN, given the covariates $X = x$, tries to distinguish between generated counterfactual data and ground truth factual data. The adversarial objective from Eq. 6 could be expanded with the law of total expectation wrt. $X$ and $A$ in the following way:

$$\mathbb{E}_{(X,A,M) \sim \mathbb{P}_{\mathrm{f}}} \left[ \log \left( D(X, \tilde{G}(X, A, M))_A \right) \right] \tag{15}$$

$$= \mathbb{E}_{X \sim \mathbb{P}(X)} \mathbb{E}_{(A,M) \sim \mathbb{P}(A,M \mid X)} \left[ \log \left( D(X, \tilde{G}(X, A, M))_A \right) \right] \tag{16}$$

$$= \mathbb{E}_{X \sim \mathbb{P}(X)} \Big[ \mathbb{E}_{M \sim \mathbb{P}(M \mid X, A=0)} \left[ \log \left( D(X, \tilde{G}(X, 0, M))_0 \right) \right] \pi_0(X) \tag{17}$$
$$+ \mathbb{E}_{(M \sim \mathbb{P}(M \mid X, A=1)} \left[ \log \left( D(X, \tilde{G}(X, 1, M))_1 \right) \right] \pi_1(X) \Big]$$

$$= \mathbb{E}_{X \sim \mathbb{P}(X)} \Big[ \mathbb{E}_{M \sim \mathbb{P}(M \mid X, A=0)} \left[ \log \left( D(X, \{M, G(X, 0, M)_1\})_0 \right) \right] \pi_0(X) \tag{18}$$
$$+ \mathbb{E}_{M \sim \mathbb{P}(M \mid X, A=1)} \left[ \log \left( 1 - D(X, \{G(X, 1, M)_0, M\})_0) \right) \right] \pi_1(X) \Big].$$

---

[6]If the conditional density of the mediator has finite values.

We give a more detailed explanation of the derivation of this step. We denote $\{M, G(X, 0, M)_1\}$ in Eq. 17 as $\tilde{G}(X, 0, M)$. We modify the output of $G$ before passing it as input to $D$. We replace the generated factual mediator $\hat{M}_A$ with the observed factual mediator $M$. We denote the new, combined data by $\tilde{G}(X, A, M)$. $\tilde{G}(X, 0, M)$ means the input of the discriminator for $A = 0$. In Eq. 4, we have defined $\tilde{G}(X, A, M)_a$ via

$$\tilde{G}(X, A, M)_a = \begin{cases} M, & \text{if } A = a \\ G(X, A, M)_a, & \text{if } A = a' \end{cases} \tag{19}$$

For the first term in Eq. 17, the mediator $M$ is drawn from $\mathbb{P}(M \mid X, A = 0)$. For $A = 0$, we have

$$\tilde{G}(X, 0, M)_a = \begin{cases} M, & \text{if } A = a \\ G(X, 0, M)_a, & \text{if } A = a' \end{cases} \tag{20}$$

When $a = 0$, we have $\tilde{G}(X, 0, M)_0 = M$; when $a = 1$, we have $\tilde{G}(X, 0, M)_1 = G(X, 0, M)_1$. Therefore, we can write $\tilde{G}(X, 0, M) = \{M, G(X, 0, M)_1\}$. By replacing this term, we have shown the first term in Eq. 17 equals the first term in Eq. 18.

Similarly, for $A = 1$, we have $\tilde{G}(X, 1, M) = \{G(X, 1, M)_0, M\}$, because, when $a = 0$ and $A = a'$, we have $\tilde{G}(X, 1, M)_0 = G(X, 1, M)_0$; when $a = 1$, we have $\tilde{G}(X, 0, M)_1 = M$.

The discriminator $D$ then determines which component of $\tilde{G}$ is the observed factual mediator and thus outputs the corresponding probability, which is given by Eq. 5, i.e., $D(X, \tilde{G})_a = \hat{\mathbb{P}}(M = \tilde{G}_a \mid X, \tilde{G}) = \hat{\mathbb{P}}(A = a \mid X, \tilde{G})$. As it is the corresponding probability, therefore, the sum of the $D(X, \tilde{G})_0$ and $D(X, \tilde{G})_1$ should be 1. By replacing the term $\tilde{G}(X, 1, M) = \{G(X, 1, M)_0, M\}$ as we have shown above, we have

$$\log\left(D(X, \tilde{G}(X, 1, M))_1\right) = \log\left(1 - D(X, \{G(X, 1, M)_0, M\})_0\right) \tag{21}$$

Thus, we have shown that the second term in Eq. 17 equals the second term in Eq. 18.

Let $Z_0 = \{M, G(X, 0, M)_1\}$ and $Z_1 = \{G(X, 1, M)_0, M\}$ be two random variables. Then, using the law of the unconscious statistician, the expression can be converted to a weighted conditional GAN adversarial loss (Mirza & Osindero, 2014), i.e.,

$$\mathbb{E}_{X \sim \mathbb{P}(X)}\left[\mathbb{E}_{Z_0 \sim \mathbb{P}(Z_0 \mid X, A=0)}\left[\log\left(D(X, Z_0)_0\right)\right] \pi_0(X)\right. \tag{22}$$

$$\left. + \mathbb{E}_{Z_1 \sim \mathbb{P}(Z_1 \mid X, A=1)}\left[\log\left(1 - D(X, Z_1)_0\right)\right] \pi_1(X)\right]$$

$$= \mathbb{E}_{X \sim \mathbb{P}(X)}\left[\int_{\mathcal{Z}}\left(\log\left(D(X, z)_0\right)\pi_0(X)\mathbb{P}(Z_0 = z \mid X, A = 0)\right.\right. \tag{23}$$

$$\left.\left. + \log\left(1 - D(X, z)_0\right)\pi_1(X)\mathbb{P}(Z_1 = z \mid X, A = 1)\right)\mathrm{d}z\right],$$

where $\mathcal{Z} = \mathcal{M} \times \mathcal{M}$. Notably, the weights of the loss, i.e., $\pi_0(X)$ and $\pi_1(X)$, are greater than zero, due to the overlap assumption. The second term follows analogously. Following the theory from the standard GANs (Goodfellow et al., 2014b), for any $(a, b) \in \mathbb{R}^2 \setminus 0$, the function $y \mapsto \log(y)a + \log(1-y)b$ achieves its maximum in $[0, 1]$ at $\frac{a}{a+b}$. Therefore, for a given generator, an optimal discriminator is

$$D(x, z)_0 = \frac{\mathbb{P}(Z_0 = z \mid X = x, A = 0)\pi_0(x)}{\mathbb{P}(Z_0 = z \mid X = x, A = 0)\pi_0(x) + \mathbb{P}(Z_1 = z \mid X = x, A = 1)\pi_1(x)}. \tag{24}$$

Both conditional densities used in the expression above can be expressed in terms of the potential outcomes densities due to the consistency and unconfoundedness assumptions, namely

$$\mathbb{P}(Z_0 = z \mid X = x, A = 0) = \mathbb{P}(\{M = m_0, G(x, 0, M)_1 = m_1\} \mid X = x, A = 0) \tag{25}$$

$$= \mathbb{P}(\{M_0 = m_0, G(x, 0, M_0)_1 = m_1\} \mid X = x),$$

$$\mathbb{P}(Z_1 = z \mid X = x, A = 1) = \mathbb{P}(\{G(x, 1, M)_0 = m_0, M = m_1\} \mid X = x, A = 1) \tag{26}$$

$$= \mathbb{P}(\{G(x, 1, M_1)_0 = m_0, M_1 = m_1\} \mid X = x).$$

Thus, an optimal generator of the GAN then minimizes the following conditional propensity-weighted Jensen–Shannon divergence (JSD)

$$\text{JSD}_{\pi_0(x), \pi_1(x)}\Big(\mathbb{P}(\{M_0, G(x, 0, M_0)_1\} \mid X = x) \,\big|\big|\, \mathbb{P}(\{G(x, 1, M_1)_0, M_1\} \mid X = x)\Big), \quad (27)$$

where $\text{JSD}_{w_1, w_2}(\mathbb{P}_1 \,||\, \mathbb{P}_1) = w_1 \,\text{KL}(\mathbb{P}_1 \,||\, w_1 \,\mathbb{P}_1 + w_2 \,\mathbb{P}_2) + w_2 \,\text{KL}(\mathbb{P}_2 \,||\, w_1 \,\mathbb{P}_1 + w_2 \,\mathbb{P}_2)$ and where $\text{KL}(\mathbb{P}_1 \,||\, \mathbb{P}_1)$ is Kullback–Leibler divergence. The Jensen–Shannon divergence is minimized, when $G(x, 0, M_0)_1 = M_1$ and $G(x, 1, M_1)_0 = M_0$ conditioned on $X = x$ (in distribution), since, in this case, it equals to zero, i.e.,

$$\mathbb{P}(G(x, a, M_a)_{a'} \mid X = x) = \mathbb{P}(M_{a'} \mid X = x). \quad (28)$$

Finally, due to the unconfoundedness assumption, the generator of our GAN estimates the potential mediator distributions with counterfactual sensitive attributes, i.e.,

$$\mathbb{P}(G(x, a, M_a)_{a'} \mid X = x, A = a) = \mathbb{P}(M_{a'} \mid X = x, A = a) \quad (29)$$

in distribution.

(ii) For a given factual observation, $X = x, A = a, M = m$, our generator yields a deterministic output, i.e.,

$$\mathbb{P}(G(x, a, M_a)_{a'} \mid X = x, A = a, M = m) = \mathbb{P}(G(x, a, m)_{a'} \mid X = x, A = a, M = m) \quad (30)$$
$$= \delta(G(x, a, m)_{a'}). \quad (31)$$

At the same time, this counterfactual distribution is connected with the potential mediators' distributions with counterfactual sensitive attributes, $\mathbb{P}(M_{a'} = m' \mid X = x, A = a)$, via the law of total probability:

$$\mathbb{P}(M_{a'} = m' \mid X = x, A = a) = \mathbb{P}(G(x, a, M)_{a'} = m' \mid X = x, A = a) \quad (32)$$

$$= \int_{\mathcal{M}} \delta(G(x, a, m)_{a'} - m') \,\mathbb{P}(M = m \mid X = x, A = a) \,\mathrm{d}m \quad (33)$$

$$= \sum_{m:\, G(x, a, m)_{a'} = m'} |\nabla_m G(x, a, m)_{a'}|^{-1} \,\mathbb{P}(M = m \mid X = x, A = a). \quad (34)$$

Due to the unconfoundedness and the consistency assumptions, this is equivalent to

$$\mathbb{P}(M = m' \mid X = x, A = a') = \sum_{m:\, G(x, a, m)_{a'} = m'} |\nabla_m G(x, a, m)_{a'}|^{-1} \,\mathbb{P}(M = m \mid X = x, A = a). \quad (35)$$

The equation above has only two solutions wrt. $G(x, a, \cdot)$ in the class of the continuously differentiable functions (Corollary 3 in (Melnychuk et al., 2023)), namely:[7]

$$G(x, a, m)_{a'} = \mathbb{F}^{-1}(\pm \mathbb{F}(m; x, a) \mp 0.5 + 0.5; x, a'), \quad (36)$$

where $\mathbb{F}(\cdot; x, a)$ and $\mathbb{F}^{-1}(\cdot; x, a)$ are a CDF and an inverse CDF of $\mathbb{P}(M \mid X = x, A = a)$. Thus, the generator of GAN exactly matches one of the two BGM solutions from (i). This concludes that our generator consistently estimates the counterfactual distribution of the mediator, $\mathbb{P}(M_{a'} \mid X = x, A = a, M = m)$, up to a measure-preserving indeterminacy. $\qquad\square$

**Corollary 1.** *The results of the Lemma 3 naturally generalize to sensitive attributes with more categories, i.e., $\mathcal{A} = \{0, 1, \ldots, k - 1\}, k > 2$.*

*Proof.* We want to show that, when $\mathcal{A} = \{0, 1, \ldots, k - 1\}, k > 2$, the generator is still able to learn the potential mediator distributions with the counterfactual distributions. For that, we follow same

---

[7]Under mild conditions, the counterfactual distributions cannot be defined via the point mass distribution with non-monotonous functions, even if we assume the extension of BGMs to all non-monotonous continuously differentiable functions.

derivation steps, as in part (i) of the proof of Lemma 3. This brings us to the following equality for the loss of the discriminator:

$$\mathbb{E}_{(X,A,M)\sim\mathbb{P}_{\mathrm{f}}}\left[\log\left(D(X,\tilde{G}(X,A,M))_A)\right)\right] \tag{37}$$

$$=\mathbb{E}_{X\sim\mathbb{P}(X)}\Bigg[\int_{\mathcal{Z}}\Big(\log\left(D(X,z)_0\right)\pi_0(X)\mathbb{P}(Z_0=z\mid X,A=0) \tag{38}$$

$$+\log\left(D(X,z)_1\right)\pi_1(X)\mathbb{P}(Z_1=z\mid X,A=1) \tag{39}$$

$$\dots \tag{40}$$

$$+\log\left(D(X,z)_{k-2}\right)\pi_{k-2}(X)\mathbb{P}(Z_{k-2}=z\mid X,A=k-2) \tag{41}$$

$$+\log\Big(1-\sum_{j=0}^{k-2}D(X,z)_j\Big)\pi_{k-1}(X)\mathbb{P}(Z_{k-1}=z\mid X,A=k-1)\Big)\,\mathrm{d}z\Bigg], \tag{42}$$

where

$$Z_0 = \{M, G(X,0,M)_1, G(X,0,M)_2, \dots, G(X,0,M)_{k-1})\}, \tag{43}$$

$$Z_1 = \{G(X,1,M)_0, M, G(X,1,M)_2, \dots, G(X,1,M)_{k-1}\}, \tag{44}$$

$$\dots \tag{45}$$

$$Z_{k-1} = \{G(X,k-1,M)_0, G(X,k-1,M)_1, \dots, M\}. \tag{46}$$

Then, it is easy to see that, for a given generator, an optimal discriminator is (analogously to Eq. 24)

$$D(x,z)_a = \frac{\mathbb{P}(Z_a=z\mid X=x,A=a)\,\pi_a(x)}{\sum_{j=0}^{k-1}\mathbb{P}(Z_j=z\mid X=x,A=j)\,\pi_j(x)}\quad\text{for all }a\in\mathcal{A}. \tag{47}$$

This happens, as, for any $(a_0,\dots,a_{k-1})\in\mathbb{R}^k\setminus 0$, the function $(y_0,y_1,\dots y_{k-2})\mapsto\log(y_0)a_0 + \log(y_1)a_1+\dots+\log(y_{k-2})a_{k-2}+\log(1-\sum_{j=0}^{k-2}y_j)a_{k-1}$ achieves its maximum in $[0,1]$ at $\left(\frac{a_0}{\sum_{j=0}^{k-1}a_j}, \frac{a_1}{\sum_{j=0}^{k-1}a_j}, \dots, \frac{a_{k-2}}{\sum_{j=0}^{k-1}a_j}\right)$. Then, an optimal generator of the GAN aims to minimize the propensity-weighted multi-distribution JSD, i.e.,

$$\mathrm{JSD}_{\pi_0(x),\pi_1(x),\dots,\pi_{k-1}(x)}\Big(\mathbb{P}(\{M_0, G(x,0,M_0)_1, G(x,0,M_0)_2, \dots, G(x,0,M_0)_{k-1}\}\mid X=x),$$

$$\mathbb{P}(\{G(x,1,M_1)_0, M_1, G(x,1,M_1)_2, \dots, G(x,1,M_1)_{k-1}\}\mid X=x),$$

$$\dots$$

$$\mathbb{P}(\{G(x,k-1,M_{k-1})_0, G(x,k-1,M_{k-1})_1, \dots, M_{k-1}\}\mid X=x)\Big). \tag{48}$$

The JSD is minimized, when all the distributions are equal. If we additionally look at the marginalized distributions, the following equalities will hold

$$\mathbb{P}(G(x,a,M_a)_{a'}\mid X=x) = \mathbb{P}(M_{a'}\mid X=x)\quad\text{for all }a\neq a'\in\mathcal{A}. \tag{49}$$

This concludes the proof of the Corollary, as all additional steps are analogous to the Lemma 3. $\square$

**Remark 3.** *We proved that the generator converges to one of the two BGM solutions in Eq. 14. Which solution the generator exactly returns depends on the initialization and the optimizer. Notably, the difference between the two solutions is negligibly small, when the variability of the mediator is low. To demonstrate this, we assume (without the loss of generality) that the ground-truth counterfactual mediator follows one of the BGM solutions, e.g.,* $\mathbb{P}(M_{a'}\mid X=x,A=a,M=m) = \delta(\mathbb{F}^{-1}(\mathbb{F}(m;x,a);x,a'))$*; and our GAN estimates another, i.e.,* $\mathbb{P}(M_{a'}\mid X=x,A=a,M=m) = \delta(\mathbb{F}^{-1}(1-\mathbb{F}(m;x,a);x,a'))$*. Then, assuming a perfect fit of the GAN, the conditional expectation*

*of the squared difference between ground-truth counterfactual mediator and estimated mediator is*

$$\sup_{G \in \mathcal{G}} \mathbb{E}\left[\left\|M_{A'} - \hat{M}_{A'}\right\|_2^2 \mid X = x, A = a\right] \tag{50}$$

$$= \mathbb{E}\left[\left|\mathbb{F}^{-1}(\mathbb{F}(M; x, a); x, a') - \mathbb{F}^{-1}(1 - \mathbb{F}(M; x, a); x, a')\right| \mid X = x, A = a\right] \tag{51}$$

$$= \mathbb{E}\left[\left|\mathbb{F}^{-1}(U; x, a') - \mathbb{F}^{-1}(1 - U; x, a')\right|\right] \tag{52}$$

$$= \int_0^1 \left|\mathbb{F}^{-1}(u; x, a') - \mathbb{F}^{-1}(1 - u; x, a')\right| \mathrm{d}u \tag{53}$$

$$\leq \int_0^1 \left|\mathbb{F}^{-1}(u; x, a') - \mu(x, a')\right| \mathrm{d}u + \int_0^1 \left|\mathbb{F}^{-1}(1 - u; x, a') - \mu(x, a')\right| \mathrm{d}u \tag{54}$$

$$= 2\,\mathbb{E}\left[\left|M - \mu(x, a')\right| \mid X = x, A = a'\right] \tag{55}$$

$$\overset{(*)}{\leq} 2\sqrt{\mathrm{Var}\left[M \mid X = x, A = a'\right]}, \tag{56}$$

*where* $(*)$ *holds as an inequality between the mean absolute deviation and the standard deviation. This result also holds for high-dimensional mediators, where there is a continuum of solutions in the class of continuously differentiable functions (Chen & Gopinath, 2000). Thus, if the conditional standard deviation of the mediator is high, a combination of multiple GANs is used to enforce the worst-case counterfactual fairness.*

### F.2 PROOF OF LEMMA 2

Here, we prove Lemma 2 from the main paper, which states that our counterfactual regularization achieves counterfactual fairness if our generator consistently estimates the counterfactuals.

**Lemma 4** (Counterfactual mediator regularization bound). *Given the prediction model $h$ that is Lipschitz continuous with a Lipschitz constant $\mathcal{C}$, we have*

$$\mathbb{E}\left[\left\|(h(X, M) - h(X, M_{A'})\right\|_2^2\right] \leq \mathcal{C}\,\mathbb{E}\left[\left\|M_{A'} - \hat{M}_{A'}\right\|_2^2\right] + \sup_{G \in \mathcal{G}} \mathcal{R}_{\mathrm{cm}}(h, G), \text{ for every } G \in \mathcal{G}, \tag{57}$$

*where* $\hat{M}_{A'} = G(X, A, M)_{A'}$ *and* $\mathcal{G}$ *is a set of all the generators, minimizing the Eq. 7.*

*Proof.* Using triangle inequality, we yield

$$\mathbb{E}\left[\|h(X, M) - h(X, M_{A'})\|_2^2\right] \tag{58}$$

$$= \mathbb{E}\left[\left\|h(X, M) - h(X, M_{A'}) + h(X, \hat{M}_{A'}) - h(X, \hat{M}_{A'})\right\|_2^2\right] \tag{59}$$

$$\leq \mathbb{E}\left[\left\|h(X, M) - h(X, \hat{M}_{A'})\right\|_2^2\right] + \mathbb{E}\left[\left\|h(X, \hat{M}_{A'}) - h(X, M_{A'})\right\|_2^2\right] \tag{60}$$

$$= \mathbb{E}\left[\left\|h(X, \hat{M}_{A'}) - h(X, M_{A'})\right\|_2^2\right] + \mathcal{R}_{\mathrm{cm}}(h, G) \tag{61}$$

$$\leq \mathcal{C}\,\mathbb{E}\left[\left\|(X, \hat{M}_{A'}) - (X, M_{A'})\right\|_2^2\right] + \mathcal{R}_{\mathrm{cm}}(h, G) \tag{62}$$

$$= \mathcal{C}\,\mathbb{E}\left[\left\|M_{A'} - \hat{M}_{A'}\right\|_2^2\right] + \mathcal{R}_{\mathrm{cm}}(h, G) \tag{63}$$

$$\leq \mathcal{C}\,\mathbb{E}\left[\left\|M_{A'} - \hat{M}_{A'}\right\|_2^2\right] + \sup_{G \in \mathcal{G}} \mathcal{R}_{\mathrm{cm}}(h, G). \tag{64}$$

$\square$

# G  TRAINING ALGORITHM OF GCFN

---

**Algorithm 1** Training algorithm of GCFN

---

1: **Input.** Training dataset $\mathcal{D}$; fairness weight $\lambda$; number of training GANs to train $s$; number of training epoch for each GAN $e_1$; number of training prediction model epoch $e_2$; minibatch of size $n$; training supervised loss weight $\alpha$

2: **Init.** Generator $G$ parameters: $\theta_g$; discriminator $D$ parameters: $\theta_d$; prediction model $h$ parameters: $\theta_h$

3: *Step 1: Training GAN to learn to generate counterfactual mediator*

4: **for** $j \in \{1, \dots, s\}$ **do**

5:     **for** $e_1$ **do**

6:         **for** $k$ steps **do**

7:             Sample minibatch of $n$ examples $\left\{x^{(i)}, a^{(i)}, m^{(i)}\right\}_{i=1}^n$ from $\mathcal{D}$

8:             Compute generator output $G_j\left(x^{(i)}, a^{(i)}, m^{(i)}\right)_a = \hat{m}_a^{(i)}$   for $a \in \{0, 1\}$

9:             Modify $G_j$ output to $\tilde{G}_{j a}^{(i)} = \begin{cases} m^{(i)}, & \text{if } a^{(i)} = a, \\ \hat{m}_a^{(i)}, & \text{if } a^{(i)} = a' \end{cases}$   for $a \in \{0, 1\}$

10:            Update the discriminator via stochastic gradient ascent

$$\nabla_{\theta_d} \frac{1}{n} \sum_{i=1}^n \left[\log\left(D_j(x^{(i)}, \tilde{G}_j^{(i)})_{a^{(i)}}\right)\right]$$

11:         **end for**

12:         **for** $k$ steps **do**

13:             Sample minibatch of $n$ examples $\left\{x^{(i)}, a^{(i)}, m^{(i)}\right\}_{i=1}^n$ from $\mathcal{D}$

14:             Compute generator output $G_j\left(x^{(i)}, a^{(i)}, m^{(i)}\right)_a = \hat{m}_a^{(i)}$   for $a \in \{0, 1\}$

15:             Modify $G_j$ output to $\tilde{G}_{j a}^{(i)} = \begin{cases} m^{(i)}, & \text{if } a^{(i)} = a, \\ \hat{m}_a^{(i)}, & \text{if } a^{(i)} = a' \end{cases}$   for $a \in \{0, 1\}$

16:            Update the generator via stochastic gradient descent

$$\nabla_{\theta_g} \frac{1}{n} \sum_{i=1}^n \left[\log\left(D_j(x^{(i)}, \tilde{G}_j^{(i)})_{a^{(i)}}\right) + \log\left(1 - D_j(x^{(i)}, \tilde{G}_j^{(i)})_{1-a^{(i)}}\right) + \alpha \left\|m^{(i)} - G_j\left(x^{(i)}, a^{(i)}, m^{(i)}\right)_{a^{(i)}}\right\|_2^2\right]$$

17:         **end for**

18:     **end for**

19: **end for**

20: *Step 2: Training prediction model with counterfactual mediator regularization*

21: **for** $e_2$ **do**

22:     Sample minibatch of $n$ examples $\left\{x^{(i)}, a^{(i)}, m^{(i)}, y^{(i)}\right\}_{i=1}^n$ from $\mathcal{D}$

23:     Generate $\hat{m}_j^{(i)}$ from $G_j\left(x^{(i)}, a^{(i)}, m^{(i)}\right)$

24:     Compute counterfactual mediator regularization

$$\mathcal{R}_{\mathrm{cm}}(h, G_j) = \left\|h(x^{(i)}, m^{(i)}) - h(x^{(i)}, \hat{m}_{j, a'^{(i)}}^{(i)})\right\|_2^2$$

25:     Update the prediction model via stochastic gradient descent

$$\nabla_{\theta_h} \frac{1}{n} \sum_{i=1}^n \left[y^{(i)} \log\left(h(x^{(i)}, m^{(i)})\right) + (1 - y^{(i)}) \log\left(1 - h(x^{(i)}, m^{(i)})\right) + \lambda \max_{j=1}^s \mathcal{R}_{\mathrm{cm}}(h, G_j)\right]$$

26: **end for**

27: **Output.** Counterfactually fair prediction model $h$

---

# H   GENERALIZATION TO MULTIPLE SOCIAL GROUPS

## H.1   THEORETICAL INSIGHTS

We provide proof that our method can naturally generalize to sensitive attributes with more categories in Appendix F, Corollary 1.

## H.2   STEP 1: GAN FOR GENERATING COUNTERFACTUAL OF THE MEDIATOR

Our method can easily extended to scenarios with multiple social groups. Suppose we have $k$ categories, then the sensitive attribute $A \in \mathcal{A}$, where $\mathcal{A} = \{0, 1, \ldots, k-1\}$ and $k > 2$.

The output of the generator $G$ is $k$ potential mediators, i.e., $\hat{M}_0, \hat{M}_1, \ldots, \hat{M}_{k-1}$, from which one is factual and the others are counterfactual.

$$G(X, A, M)_a = \hat{M}_a \quad \text{for} \quad a \in \{0, 1, ..., k-1\} \tag{65}$$

The reconstruction loss of the generator is the same as the binary case,

$$\mathcal{L}_{\mathrm{f}}(G) = \mathbb{E}_{(X,A,M) \sim \mathbb{P}_{\mathrm{f}}} \left[ \|M - G(X, A, M)_A\|_2^2 \right], \tag{66}$$

where $\| \cdot \|_2$ is the $L_2$-norm.

The discriminator $D$ is designed to differentiate the factual mediator $M$ (as observed in the data) from the $k-1$ generated counterfactual mediators (as generated by $G$).

We modify the output of $G$ before passing it as input to $D$: We replace the generated factual mediator $\hat{M}_A$ with the observed factual mediator $M$. We denote the new, combined data by $\tilde{G}(X, A, M)$, which is defined via

$$\tilde{G}(X, A, M)_a = \begin{cases} M, & \text{if } A = a, \\ G(X, A, M)_a, & \text{Otherwise}, \end{cases} \quad \text{for } a \in \{0, 1, ..., k-1\}. \tag{67}$$

The discriminator $D$ then determines which component of $\tilde{G}$ is the observed factual mediator and thus outputs the corresponding probability. Formally, for the input $(X, \tilde{G})$, the output of the discriminator $D$ is

$$D\left(X, \tilde{G}\right)_a = \hat{\mathbb{P}}\left(M = \tilde{G}_a \mid X, \tilde{G}\right) = \hat{\mathbb{P}}\left(A = a \mid X, \tilde{G}\right) \quad \text{for } a \in \{0, 1, ..., k-1\}. \tag{68}$$

Our GAN is trained in an adversarial manner: (i) the generator $G$ seeks to generate counterfactual mediators in a way that minimizes the probability that the discriminator can differentiate between factual mediators and counterfactual mediators, while (ii) the discriminator $D$ seeks to maximize the probability of correctly identifying the factual mediator. We thus use an adversarial loss $\mathcal{L}_{\mathrm{adv}}$ by

$$\mathcal{L}_{\mathrm{adv}}(G, D) = \mathbb{E}_{(X,A,M) \sim \mathbb{P}_{\mathrm{f}}} \left[ \log\left(D(X, \tilde{G}(X, A, M))_A\right) \right]. \tag{69}$$

Overall, our GAN is trained through an adversarial training procedure with a minimax problem as

$$\min_G \max_D \mathcal{L}_{\mathrm{adv}}(G, D) + \alpha \mathcal{L}_{\mathrm{f}}(G), \tag{70}$$

with a hyperparameter $\alpha$ on $\mathcal{L}_{\mathrm{f}}$.

## H.3   STEP 2: COUNTERFACTUAL FAIR PREDICTION THROUGH COUNTERFACTUAL MEDIATOR REGULARIZATION

We use the output of the GAN to train a prediction model $h$ under counterfactual fairness in a supervised way. Our counterfactual mediator regularization $\mathcal{R}_{\mathrm{cm}}(h, G)$ thus is

$$\mathcal{R}_{\mathrm{cm}}(h, G) = \mathbb{E}_{(X,A,M) \sim \mathbb{P}_{\mathrm{f}}} \left[ \frac{1}{(k-1)} \sum_{\substack{a=0 \\ a \neq A}}^{k-1} \left\| h(X, M) - h\left(X, \hat{M}_a\right) \right\|_2^2 \right]. \tag{71}$$

The training loss is

$$\mathcal{L}(h) = \mathcal{L}_{\mathrm{ce}}(h) + \lambda \sup_{G \in \mathcal{G}} \mathcal{R}_{\mathrm{cm}}(h, G). \tag{72}$$

# I  DATASET

## I.1  SYNTHETIC DATA

Analogous to prior works that simulate synthetic data for benchmarking (Kim et al., 2021; Kusner et al., 2017; Quinzan et al., 2022), we generate our synthetic dataset in the following way. The covariates $X$ is drawn from a standard normal distribution $\mathcal{N}(0,1)$. The sensitive attribute $A$ follows a Bernoulli distribution with probability $p$, determined by a sigmoid function $\sigma$ of $X$ and a Gaussian noise term $U_A$. We then generate the mediator $M$ as a function of $X$, $A$, and a Gaussian noise term $U_M$. Finally, the target $Y$ follows a Bernoulli distribution with probability $p_y$, calculated by a sigmoid function of $X$, $M$, and a Gaussian noise term $U_Y$. $\beta_i$ ($i \in [1,6]$) are the coefficients. Let $\sigma(x) = \frac{1}{1+e^{-x}}$ represent the sigmoid function. Formally, we yield

$$\begin{cases} X = U_X & U_X \sim \mathcal{N}(0,1) \\ A \sim \text{Bernoulli}\left(\sigma(\beta_1 X + U_A)\right) & U_A \sim \mathcal{N}(0, 0.01) \\ M = \beta_2 X + \beta_3 A + U_M & U_M \sim \mathcal{N}(0, 0.01) \\ Y \sim \text{Bernoulli}\left(\sigma(\beta_5 X + \beta_6 M + U_Y)\right) & U_Y \sim \mathcal{N}(0, 0.01) \end{cases} \tag{73}$$

We sample 10,000 observations and use 20% as the test set.

## I.2  SEMI-SYNTHETIC DATA

**LSAC dataset.** The Law School (LSAC) dataset (Wightman, 1998) contains information about the law school admission records. We use the LSAC dataset to construct two semi-synthetic datasets. In both, we set the sensitive attribute to *gender*. We take *resident* and *race* from the LSAC dataset as confounding variables. The *LSAT* and *GPA* are the mediator variables, and the *admissions decision* is our target variable. We simulate 101,570 samples and use 20% as the test set. We denote $M_1$ as GPA score, $M_2$ as LSAT score, $X_1$ as resident, and $X_2$ as race. Further, $w_{X_1}, w_{X_2}, w_A, w_{M_1}, w_{M_2}$ are the coefficients. $U_{M_1}, U_{M_2}, U_Y$ are the Gaussian noise. Let $\sigma(x) = \frac{1}{1+e^{-x}}$ represent the sigmoid function.

We follow the prior work (Bica et al., 2020) to produce two different semi-synthetic datasets as follows. For the first one, we use the sigmoid function on linear combinations and for the second one, we use the sinus function that could make extrapolation more challenging for our GCFN.

■ Semi-synthetic dataset "sigmoid":

$$\begin{cases} M_1 = w_{M_1}\left(\sigma(w_A A + w_{X_1} X_1 + w_{X_2} X_2 + U_{M_1})\right) & U_{M_1} \sim \mathcal{N}(0, 0.01) \\ M_2 = w_{M_2} + w_{M_1}\left(\sigma(w_A S + w_{X_1} X_1 + w_{X_2} X_2 + U_{M_2})\right) & U_{M_2} \sim \mathcal{N}(0, 0.01) \\ Y \sim \text{Bernoulli}\left(\sigma(w_{M_1} M_1 + w_{M_2} M_2 + w_{X_1} X_1 + w_{X_2} X_2 + U_Y)\right) & U_Y \sim \mathcal{N}(0, 0.01) \end{cases}$$
$$\tag{74}$$

■ Semi-synthetic "sin":

$$\begin{cases} M_1 = w_A \cdot A - \sin\left(\pi \times (w_{X_1} X_1 + w_{X_2} X_2 + U_{M_1})\right) & U_{M_1} \sim \mathcal{N}(0, 0.01) \\ M_2 = w_A \cdot A - \sin\left(\pi \times (w_{X_1} X_1 + w_{X_2} X_2 + U_{M_2})\right) & U_{M_2} \sim \mathcal{N}(0, 0.01) \\ Y \sim \text{Bernoulli}\left(\sigma(w_{M_1} M_1 + w_{M_2} M_2 + w_{X_1} X_1 + w_{X_2} X_2 + U_Y)\right) & U_Y \sim \mathcal{N}(0, 0.01) \end{cases}$$
$$\tag{75}$$

## I.3  REAL-WORLD DATA

**UCI Adult dataset:** The UCI Adult dataset (Asuncion & Newman, 2007) captures information about 48,842 individuals including their sociodemographics. Our aim is to predict if individuals earn more than USD 50k per year. We follow the setting of earlier research (Kim et al., 2021; Nabi & Shpitser, 2018; Quinzan et al., 2022; Xu et al., 2019). We treat *gender* as the sensitive attribute and set mediator variables to be *marital status*, *education level*, *occupation*, *hours per week*, and *work class*. The causal graph of the UCI dataset is in Fig. 8. We take 20% as the test set.

**COMPAS dataset:** COMPAS (Correctional Offender Management Profiling for Alternative Sanctions) (Angwin et al., 2016) was developed as a decision support tool to score the likelihood of a

person's recidivism. The score ranges from 1 (lowest risk) to 10 (highest risk). The dataset further contains information about whether there was an actual recidivism (reoffended) record within 2 years after the decision. Overall, the dataset has information about over 10,000 criminal defendants in Broward County, Florida. We treat *race* as the sensitive attribute. The mediator variables are the features related to prior convictions and current charge degree. The target variable is the *recidivism* for each defendant. The causal graph of the COMPAS dataset is in Fig. 9. We take 20% as test set.

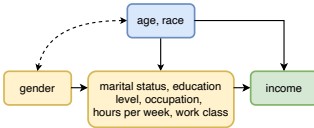

Figure 8: Causal graph of UCI dataset.

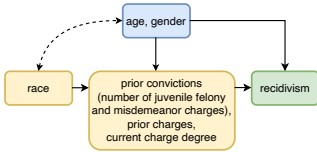

Figure 9: Causal graph of COMPAS dataset.

## J  IMPLEMENTATION DETAILS

### J.1  IMPLEMENTATION OF OUR METHOD

To ensure our GCFN to achieve counterfactual fairness, we train $s = 10$ GANs and consider the worst-case counterfactual fairness, i.e., we take the maximum value of counterfactual mediator regularization $\max_{j=1}^{s} \mathcal{R}_{\text{cm}}(h, G_j)$. Our GCFN is implemented in PyTorch. Both the generator and the discriminator in the GAN model are designed as deep neural networks, each with a hidden layer of dimension $64$. LeakyReLU is employed as the activation function and batch normalization is applied in the generator to enhance training stability. The GAN training procedure is performed for $300$ epochs with a batch size of $256$ at each iteration. We set the learning rate to $0.0005$. Following the GAN training, the prediction model, structured as a multilayer perceptron (MLP), is trained separately. This classifier can incorporate spectral normalization in its linear layers to ensure Lipschitz continuously. It is trained for $30$ epochs, with the same learning rate of $0.005$ applied. The training time of our GCFN on (semi-) synthetic dataset is comparable to or smaller than the baselines.

### J.2  IMPLEMENTATION OF BENCHMARKS

We implement **CFAN** (Kusner et al., 2017) in PyTorch based on the paper's source code in R and Stan on `https://github.com/mkusner/counterfactual-fairness`. We use a VAE to infer the latent variables. For **mCEVAE** (Pfohl et al., 2019), we follow the implementation from `https://github.com/HyemiK1m/DCEVAE/tree/master/Tabular/mCEVAE_baseline`. We implement **CFGAN** (Xu et al., 2019) in PyTorch based on the code of (Abroshan et al., 2022) and the TensorFlow source code of (Xu et al., 2019). We implement **ADVAE** (Grari et al., 2023) in PyTorch. For **DCEVAE** (Kim et al., 2021), we use the source code of the author of **DCEVAE** (Kim et al., 2021). We use **HSCIC** (Quinzan et al., 2022) source implementation from the supplementary material provided on the OpenReview website `https://openreview.net/forum?id=ERjQnrmLKH4`. We performed rigorous hyperparameter tuning for all baselines.

### J.3  HYPERPARAMETER TUNING.

We perform a rigorous procedure to optimize the hyperparameters for the different methods as follows. For DCEVAE (Kim et al., 2021) and mCEVAE (Pfohl et al., 2019), we follow the hyperparameter optimization as described in the supplement of (Kim et al., 2021). For ADVAE (Grari et al., 2023) and CFGAN (Xu et al., 2019), we follow the hyperparameter optimization as described in their paper. For both HSCIC and our GCFN, we have an additional weight that introduces a trade-off between accuracy and fairness. This provides additional flexibility to decision-makers as they tailor the methods based on the fairness needs in practice (Quinzan et al., 2022). We then benchmark the utility of different methods across different choices of $\gamma$ of the utility function in Sec. 5.1. This allows us thus to optimize the trade-off weight $\lambda$ inside HSCIC and our GCFN using grid search. For HSCIC, we experiment with $\lambda = 0.1, 0.5, 1, 5, 10, 15, 20$ and choose the best for them across different datasets. For our method, we experiment with $\lambda = 0.1, 0.5, 1, 1.5, 2$. Since the utility function considers two metrics, across the experiments on (semi-)synthetic dataset, the weight $\lambda$ is set to $0.5$ to get a good balance for our method.

# K   ADDITIONAL EXPERIMENTAL RESULTS

## K.1   RESULTS FOR SYNTHETIC DATASET

**Setting:** We follow previous works that simulate a fully synthetic dataset for performance evaluations (Kim et al., 2021; Quinzan et al., 2022). The details of data generation process are in Appendix I. We generate 10,000 samples and use 20% as the test set.

**Results:** Results are shown in Fig. 10. We again find that our method is highly effective.

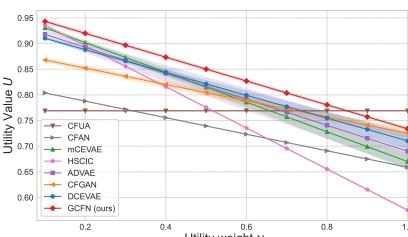

Figure 10: Results for synthetic datasets. A larger utility is better. Shown: mean $\pm$ std over 5 runs.

## K.2   ADDITIONAL INSIGHTS: WHY OUR METHOD DOES NOT SIMPLY REPRODUCE THE FACTUAL MEDIATORS BUT ACTUALLY LEARNS THE COUNTERFACTUAL MEDIATORS?

As an additional analysis, we now provide further insights into how our GCFN operates. Specifically, one may think that our GCFN simply learns to reproduce factual mediators in the GAN rather than actually learning the counterfactual mediators. However, this is *not* the case. To show this, we compare the (1) the factual mediator $M$, (2) the ground-truth counterfactual mediator $M_{A'}$, and (3) the generated counterfactual mediator $\hat{M}_{A'}$. The normalized mean squared error (MSE) between them is in Table 2 ($D_1$, $D_2$, $D_3$ refer to synthetic, semi-syn (sigmoid) and semi-syn (sin) dataset in Appendix I, respectively. We find: (1) The factual mediator and the generated counterfactual mediator are highly dissimilar. This is shown by a normalized $\text{MSE}(M, \hat{M}_{A'})$ of $\approx 1$. (2) The ground-truth counterfactual mediator and our generated counterfactual mediator are highly similar. This shown by a normalized $\text{MSE}(M_{A'}, \hat{M}_{A'})$ of close to zero. In sum, our GCFN is effective in learning counterfactual mediators (and does *not* reproduce the factual data).

We further give explanations for why our GAN does not copy the factual values but learns the counterfactual values (due to its custom design!). It is true that during training, we cannot directly learn the counterfactual mediators in a supervised way as they are unobservable. Instead, we can only leverage the reconstruction loss on the factual mediators. The reason why we can still learn the correct counterfactual mediators is due to the adversarial training process of the generator. By training the discriminator to differentiate between factual and generated counterfactual mediators, the generator is guided to learn the correct counterfactual distribution. It could be seen as a form of teacher forcing.

Importantly, there are three important arguments for why our method does not simply reproduce the factual mediators but actually learns the counterfactual mediators even though it is not observed.

**Intuitively:** The input of the discriminator $D$ contains the counterfactual and the factual and the order of them is intentionally randomized. Suppose, hypothetically, that the factual always comes in the first place, then it is easy to distinguish. However, in the design of our framework, this is not the case. In our framework, the data are intentionally shuffled, so that factual and counterfactual positions are random.

**Technical reason:** If the generator $G$ would just copy the factual values of the mediator $M$ and output a trivial solution, it would be super hard for the discriminator $D$ to distinguish, implying that the loss of $D$ would be super large. We would observe mode collapse during training, yet which we did not observe and thus provides support for our argument.

**Theoretical reason:** We provide theoretical proof in our Lemma 1. Therein, we show theoretically that our generator consistently estimates the counterfactual distribution of the mediator $\mathbb{P}(M_{a'} \mid X = x, A = a, M = m)$. For each specific input data $(X = x, A = a, M = m)$, we then generate the

counterfactual mediator from the distribution $\mathbb{P}(M_{a'} \mid X = x, A = a, M = m)$. Hence, we offer theoretical proof that we learn counterfactual fairness correctly.

Table 2: Our GCFN can learn the distribution of the counterfactual mediator. The normalized $\text{MSE}(M_{A'}, \hat{M}_{A'})$ is $\approx 0$, showing the generated counterfactual mediator is similar to the ground-truth counterfactual mediator. In contrast, both the factual and the generated counterfactual mediator are highly dissimilar.

|  | $D_1$ | $D_2$ | $D_3$ |
|---|---|---|---|
| $\text{MSE}(M, M_{A'})$ | $1.00_{\pm 0.00}$ | $1.00_{\pm 0.00}$ | $1.00_{\pm 0.00}$ |
| $\text{MSE}(M, \hat{M}_{A'})$ | $1.21_{\pm 0.064}$ | $1.02_{\pm 0.027}$ | $1.06_{\pm 0.051}$ |
| $\text{MSE}(M_{A'}, \hat{M}_{A'})$ | $0.14_{\pm 0.052}$ | $0.05_{\pm 0.013}$ | $0.08_{\pm 0.028}$ |

$M$: ground-truth factual mediator; $M_{A'}$: ground-truth counterfactual mediator; $\hat{M}_{A'}$: generated counterfactual mediator

### K.3 COMPUTATIONAL EFFICIENCY

We initially used the approach based on several GANs to ensure that the assumptions of our theoretical foundation were met. However, this was merely done for theoretical reasons, but not for better performance. More specifically, we train several GANs to ensure the worst-case counterfactual fairness to be aligned with our theory. This gives the upper bound of the extent to which counterfactual fairness is fulfilled in the prediction model, which is needed for our theoretical guarantees. Below, we demonstrate that a single GAN is sufficient for state-of-the-art performance.

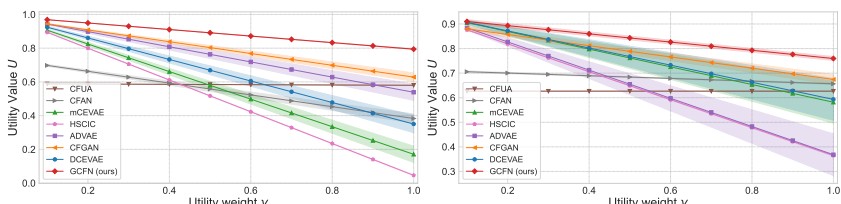

Figure 11: Results for semi-synthetic datasets with a single GAN. A larger utility is better. Shown: mean $\pm$ std over 5 runs.

We conducted the experiments on the LSAC dataset. The results are shown in Figure 11. We can see that our method based on a single GAN still gives good results and outperforms all baselines. This demonstrates the scalability of our method. In fact, in practice, the scalability of our proposed method can be simplified using a single GAN. Therefore, we recommend using a single GAN for empirical use and multiple for when theoretical guarantees are additionally needed.

### K.4 RESULTS FOR (SEMI-)SYNTHETIC DATASET

We compute the average value of the utility function $U$ over varying utility weights $\gamma \in \{0.1, \ldots, 1.0\}$ on the synthetic dataset (Fig. 12) and two different semi-synthetic datasets (Fig. 13 and Fig. 14).

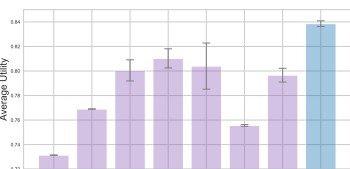

Figure 12: Average utility function value $U$ across different utility weights $\gamma$ on synthetic dataset.

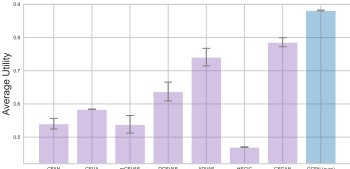

Figure 13: Average utility function value $U$ across different utility weight $\gamma$ on semi-synthetic (sigmoid) dataset.

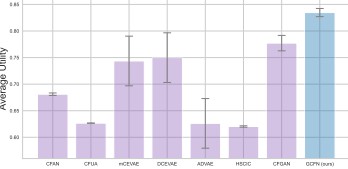

Figure 14: Average utility function value $U$ across different utility weight $\gamma$ on semi-synthetic (sin) dataset.

## K.5 RESULTS FOR UCI ADULT DATASET

We now examine the results for different fairness weights $\lambda$. For this, we report results from $\lambda = 0.5$ (Fig. 15) to $\lambda = 1000$ (Fig. 18). In line with our expectations, we see that larger values for fairness weight $\lambda$ lead the distributions of the predicted target to overlap more, implying that counterfactual fairness is enforced more strictly. This shows that our regularization $\mathcal{R}_{\mathrm{cm}}$ achieves the desired behavior.

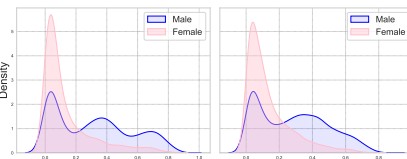

Figure 15: Density plots of the predicted target on UCI Adult dataset. Left: fairness weight $\lambda = 0$. Right: fairness weight $\lambda = 0.5$.

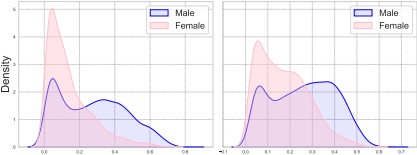

Figure 16: Density plots of the predicted target on UCI Adult dataset. Left: fairness weight $\lambda = 1$. Right: fairness weight $\lambda = 5$.

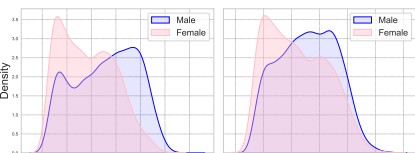

Figure 17: Density plots of the predicted target on UCI Adult dataset. Left: fairness weight $\lambda = 10$. Right: fairness weight $\lambda = 100$.

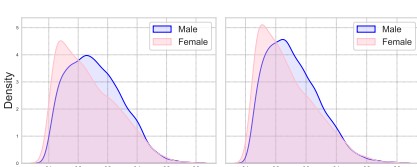

Figure 18: Density plots of the predicted target on UCI Adult dataset. Left: fairness weight $\lambda = 500$. Right: fairness weight $\lambda = 1000$.

### K.6 RESULTS FOR COMPAS DATASET

In Sec. 5, we show how black defendants are treated differently by the COMPAS score vs. our GCFN. Here, we also show how white defendants are treated differently by the COMPAS score vs. our GCFN; see Fig. 19. We make the following observations. (1) Our GCFN makes oftentimes different predictions for white defendants with a low and high COMPAS score, which is different from black defendants. (2) Our method also arrives at different predictions for white defendants with low prior charges, similar to black defendants.

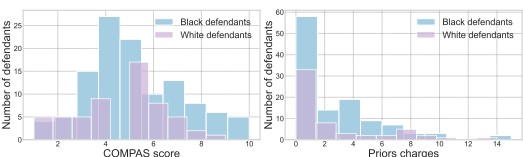

Figure 19: Distribution of white and black defendants that are treated differently using our GCFN. Left: COMPAS score. Right: Prior charges.

