# OpenReview forum: "Counterfactual fairness prediction:  Consistent estimation with generative models and theoretical guarantees"
_ICLR.cc/2025/Conference — Submitted to ICLR 2025_

### Official Review · Reviewer_Ucdx · 2024-10-17

**Soundness:** 2
**Presentation:** 2
**Contribution:** 3
**Rating:** 6
**Confidence:** 3

**Summary:**

The paper proposed a new method to produce counterfactually fair predictions. Specifically, the authors designed a GAN-like structure (GCFN) to learn the counterfactual mediator distribution when the structural causal model (SCM) satisfies certain assumptions. Then the learned mediator together with the features is used to learn the predictor. Furthermore, the paper provided some theoretical guarantees of the method. Lemma 1 claims that GCFN is able to learn the counterfactual mediator distribution only with the factual mediator values, while Lemma 2 provides a fairness guarantee. Experiments on synthetic and real datasets verify the effectiveness of the method.

**Strengths:**

1. The topic is well-motivated: learning counterfactually fair predictors is important, while VAE-based methods may produce biased estimations of latent variables.

2. Using a GAN-like structure naturally makes sense to me, and the causal structure in Figure 1 is not too restricted in my opinion.

3. The theoretical analysis provides guarantee of the proposed algorithm.

**Weaknesses:**

1.  The conditions of Lemma 1 are restrictive. Moreover, Lemma 1 seems to say we can always identify the distribution of the counterfactual mediator. Does this mean if we directly learn $P(M|X=x, A=a)$ we can learn the distribution of $M_{a'}$ and we do not necessarily need GAN? After all, the GAN can only be trained using factual data, so it approximates the factual mediator distribution.

(After reading the review of other reviewers, Lemma 1 has some problems and the authors downgrade it to a remark. I feel that the clarity of the contribution indeed has some problems. Considering many works in counterfactual fairness have restrictive assumptions, I still think this work has some merit and will retain a positive score)

2. It would be beneficial to include some examples to illustrate: (i) how VAE-based methods get a biased estimation of latent variables; (ii) What the mediators are in some practical example causal graphs.

3. The paper does not compare the proposed method with the counterfactually fair representation method [Zuo et al., 2023]. Basically, learning the counterfactual mediator also serves the function of preserving more information of the features, so it is worthwhile comparing the 2 methods.

4. It may help to include some proof intuition in the main paper.

**Questions:**

1. Could you elaborate more on Lemma 1 (see weakness 1)?

2.  Line 338:  how can we empirically measure the quality of "generated counterfactual mediator", i.e., $\|M_{A'} - \hat{M}_{A'}\|^2_2$?

---

> ### Author Response · Authors · 2024-11-20
> **Response to Reviewer Ucdx**
>
> Thank you for your review and your helpful comments! We are happy to answer your questions below, and we improved our paper as a result. We highlighted all major changes in **blue color** in our **updated PDF**.
>
> ## Response to Weaknesses
>
> **Response to W1**
>
> Thank you for asking this important question. Yes, you are right, Lemma 1 says that we can always identify the distribution of the counterfactual mediator by our tailored model and it can converge to the true counterfactual distribution. Yes, you are right that it is not necessary to have a GAN to learn the counterfactual distribution as one can also use other neural models with continuously differentiable functions (Lemma 1.1). However, **our theory serves our specific model design (Lemma 1.2)**, and **the proof is based on our tailored model architecture and properties (see Appendix D. Equation (15) to Equation (32))**, which further ensures the **theoretical guarantee for our counterfactual fairness prediction.**
>
> You are right that the GAN can only be trained with factual data as the counterfactual data is unobservable. We are happy to explain more about why our method does **not** simply reproduce the **factual** mediators but actually learns the **counterfactual** mediators. There are three important arguments for this:
>
> （1） **Intuitively**: The input of the discriminator $D$ contains the counterfactual and the factual and the _order of them is intentionally randomized_. Suppose, hypothetically, that the factual always comes in the first place, then it is easy to distinguish. However, in the design of our framework, this is **not** the case. In our framework, the data are intentionally shuffled, so that factual and counterfactual positions are random.
>
> （2）**Technical reason**: If the generator $G$ would just copy the factual of the mediator $M$ and output a trivial solution, then the same would happen as you mentioned: it would be super hard for the discriminator $D$ to distinguish both, implying that the loss of $D$ would be super large. We would thus observe mode collapse during training. Yet we did **not**observe**.
>
> （3） **Theoretical reason**: We provide theoretical proof in our Lemma 1. Therein, **we show theoretically that our generator consistently estimates the counterfactual distribution of the mediator** $\mathbb{P}(M_{a'} \mid X=x, A=a, M=m)$. For each specific input data $(X=x, A=a,M=m)$, we then generate the counterfactual mediator from the distribution $\mathbb{P}(M_{a'} \mid X=x, A=a, M=m)$. Hence, **we offer theoretical proof that we learn counterfactual values correctly**.
>
> Therefore, we performed **additional experiments** (see **Appendix K.2**). We observed that (1) The factual mediator and the generated counterfactual mediator are highly dissimilar. (2) The ground-truth counterfactual mediator and our generated counterfactual mediator are highly similar. This confirms that the mediator is not simply copying the factual values but learning the distribution of counterfactual values.
>
> **Action:** We have added **Section 5.2.2 and Appendix K.2** in our revised paper.

---

> ### Author Response · Authors · 2024-11-20
> **Response to Reviewer Ucdx**
>
> **Response to W2**
>
> Thank you for your suggestions.
>
> (i) Example to illustrate why VAE-based methods for our task can be problematic for the estimation of latent variables: The inferred latent variable should be independent of the sensitive attribute $A$ while representing all other useful information from the observation data. However, there are two main challenges: (1) The latent variable $U$ is **not** identifiable. (2) It is very hard to learn such $U$ to satisfy the above independence requirement, especially for high-dimensional or other more complicated settings. Recent research, such as [2,3], has highlighted that this choice can be **suboptimal**. The standard Gaussian prior can lead to over-regularization, contributing to poorer density estimation performance. This issue is often referred to as the posterior-collapse phenomenon, as discussed by [4].
>
>
>
> (ii) Practical example of causal graph: In our causal graph, mediators $M$ are simply all variables that can potentially be influenced by the sensitive attribute. All other variables (except for $A$ and $Y$) are modeled as covariates $X$. For example, consider a job application setting where we want to avoid discrimination by gender. Then $A$ is gender, and $Y$ is the job offer. Mediators $M$ are, for instance, education level or work experience, as both are potentially influenced by gender. In contrast, age is a covariate $X$  because it is not influenced by gender.
>
> Reference:
>
> [1] Zuo, Z., Khalili, M., & Zhang, X. (2024). Counterfactually Fair Representation. Advances in Neural Information Processing Systems, 36.
>
> [2] Takahashi, Hiroshi and Iwata, Tomoharu and Yamanaka, Yuki and Yamada, Masanori and Yagi, Satoshi. "Variational autoencoder with implicit optimal priors." In AAAI, 2019.
>
> [3] Hoffman, Matthew D., and Johnson, Matthew J. ELBO surgery: yet another way to carve up the variational evidence lower bound. In Workshop in Advances in Approximate Bayesian Inference,  NeurIPS. 2016.
>
> [4] Aaron van den Oord, Oriol Vinyals, koray kavukcuoglu. Neural Discrete Representation Learning. In  In NeurIPS.2017

---

> ### Author Response · Authors · 2024-11-20
> **Response to Reviewer Ucdx**
>
> **Response to W3**
>
> Thank you for recommending the paper [Zuo et al., 2023]. We are happy to discuss and compare this method with our method. Below, we also show the experiment results compared with this method.
>
> There are **several important differences** from our work.
>
> (1) Non-identifiability: [1] proposed a method for generating counterfactually fair representations. [1] first learns latent variables $U$ and then uses $U$ to generate multiple counterfactual samples, then generates counterfactual fair representation based on the counterfactual samples. [1] does not discuss the identifiability of counterfactuals. At the counterfactual sample generation step, there is **no** identifiability of the unobservable variable in [1], and there is **no** theoretical guarantee on the generated counterfactual samples and fair representation. This method acts as heuristics and return estimates but they do not correspond to the true value.  So, the paper may lead to predictions that are unfair. In contrast, **only our method discusses the theoretical guarantee wrt. the identifiability of counterfactuals and gives the theoretical guarantees on the generated counterfactual.**
>
> (2) Different assumptions: **[1] makes stricter assumptions**. Paper [1] separates the descendants of the sensitive attribute (which is $M$ in our paper) into $X_1$ and $X_2$, and $X_1$ is the parent of $X_2$. However, we do need to know the causal relationships of the variables inside the mediator. Hence: **the paper [1] requires more knowledge** than our settings.
>
> (3) Different methodology:  [1] consists of three steps: counterfactual sample generation; counterfactually fair representation generation; and fair model training. In contrast, our method works in two steps and is thus simpler: Step 1 uses GANs to learn the counterfactual distribution of the mediator. Step 2 uses the generated counterfactual mediators from the first step together with our counterfactual mediator regularization to enforce counterfactual fairness.
>
> (4) Different model architecture: [1] uses a VAE to infer the latent variables. We use tailored GANs to generate the counterfactual distribution of the mediator and use the adversarial training process to encourage $G$ to learn the counterfactual distribution of the mediator.
>
>
> **Experiment results: benchmarking against [1]**:
>
> We follow the original public implementation of the author of the paper [1] https://github.com/osu-srml/CF_Representation_Learning and benchmarking against [1].
> Below, we report results for utility value with different weights $\gamma \in [0.1, 0.5, 1.0]$ consistent in our paper. The experiment results on the LSAC dataset show our method has a better utility value across different weights $\gamma$. In sum, **our proposed method is better than [1] by a large margin**.
>
> | | Utility ($\gamma=0.1$) | Utility ($\gamma=0.5$) | Utility ($\gamma=1.0$) |
> |---|---|---|---|
> | paper [1] | 0.74 | 0.68 | 0.62 |
> | Ours | 0.97 | 0.89 | 0.79 |
>
> **Action:** We have added the reference of paper [1] and the comparison with this method in **our Section 2.2 and Appendix D.2**.
>
> Reference:
> [1] Zuo, Z., Khalili, M., & Zhang, X. (2024). Counterfactually Fair Representation. Advances in Neural Information Processing Systems, 36.

---

> ### Author Response · Authors · 2024-11-20
> **Response to Reviewer Ucdx**
>
> ## Response to Weaknesses
>
> **Response to W4**
>
> Thank you for your suggestions.
>
> **Action:** We have added a short proof intuition for the proofs behind Lemma 1 and Lemma 2 to our main paper while adhering to the space constraints.
>
>
> ## Answer to “Questions”
>
> **Answer to Q1:**
>
> We refer to the “Response to Weakness 1”
>
> **Answer to Q2:**
>
> We apologize that our sentence was confusing. What we mean here is that: To what extent counterfactual fairness is fulfilled in the prediction model is upper bounded by the estimation of counterfactual mediators and the counterfactual mediator regularization. The error of our generated counterfactual of the mediator is small because Lemma 1 shows that our generator consistently estimates the counterfactual distribution of the mediator.
>
> **Action:** We revised the corresponding sentence in our updated PDF.

---

> > ### Comment · Reviewer_Ucdx · 2024-11-20
> > **Thanks for your response**
> >
> > Thank you for the response and for updating the manuscript. I currently do not have other questions and will retain my score.

---

### Official Review · Reviewer_jvSK · 2024-10-26

**Soundness:** 3
**Presentation:** 2
**Contribution:** 3
**Rating:** 6
**Confidence:** 3

**Summary:**

This paper aims to address counterfactual fairness by proposing a two-step solution: (1) training a Generative Adversarial Network (GAN) to estimate counterfactual mediators and (2) training a predictor with regularization based on the estimated counterfactual mediator. Under certain assumptions, they demonstrate that their method can guarantee counterfactual fairness. They validate their method on semi-synthetic and real-world datasets.

**Strengths:**

1. This paper proposes theoretical results on counterfactual fairness. While I didn’t examine all the proofs in detail, their result seems solid.
2. The writing is clear, and the paper is easy to follow.
3. The empirical study is generally convincing.

**Weaknesses:**

**Theoretical contribution in comparison to existing works**

While I acknowledge the theoretical contribution of this paper, I believe the authors overemphasize their contribution relative to existing baselines. For example, methods in [1][2][3][4] all provide methods that satisfy counterfactual fairness under certain conditions.
1. Could the authors clarify the differences between these methods in terms of theoretical guarantees? With regard to [1], I don’t believe the primary difference lies in the need for knowledge of the ground-truth SCM, as stated in the paper. They only need to know which variables are the descendants of the sensitive attributes, which is also necessary for the method proposed in this paper. Other papers are not discussed in the current draft.
2. The authors should discuss the significance of their unique contribution more carefully. In some sections, they describe their approach as “the first neural method with theoretical guarantees,” while in others, this claim is omitted (e.g., Line 17, Line 48). This inconsistency could be misleading to readers.




**Causal Model**

Causal-based fairness notions rely heavily on the chosen causal model. Hence, it is important to clearly present the motivation behind the selected causal model. I agree that the causal model considered in this paper is general. However, could the authors provide additional insight into this model, perhaps by offering a real-world example to illustrate the role of each variable?


**Presentation**

Here are a few places where I find the presentation to be unclear or confusing
1. Line 154 - Does the theoretical result in this paper require invertibility? If so, this should be explicitly stated alongside other assumptions, as this is a fairly strong assumption.
2. In the original Standard Fairness Model, the $X$ is considered as a hidden confounder. A similar $U$ is considered in some other existing works such as [1][2]. Why can we assume access to $X$ in this paper?
3. Line 282 - Could the authors clarify why this is described as “non-trivial” and explain how it might be advantageous (if at all) compared to existing methods that require access to counterfactuals?

I am willing to adjust my score if the authors could address my concerns above.

[1] Kusner, M. J., Loftus, J., Russell, C., & Silva, R. (2017). Counterfactual fairness. Advances in neural information processing systems, 30.

[2] Wang, Y., Sridhar, D., & Blei, D. (2023). Adjusting Machine Learning Decisions for Equal Opportunity and Counterfactual Fairness. Transactions on Machine Learning Research.

[3] Zuo, Z., Khalili, M., & Zhang, X. (2023). Counterfactually fair representation. Advances in Neural Information Processing Systems, 36, 12124-12140.

[4] Zhou, Z., Liu, T., Bai, R., Gao, J., Kocaoglu, M., & Inouye, D. I. (2024). Counterfactual Fairness by Combining Factual and Counterfactual Predictions. arXiv preprint arXiv:2409.01977.

**Questions:**

Two short questions
1. Could there be an arrow from $A$ to $Y$ in Figure1?
3. Line 206 - Could the authors clarify more carefully why avoiding the abduction-action-prediction framework reduces the possibility of estimation errors?

---

> ### Author Response · Authors · 2024-11-20
> **Response to Reviewer jvSK**
>
> Thank you for your review and your helpful comments! We took all your comments at heart and improved our paper as a result. We highlighted all major changes in **blue color** in our **updated PDF**.
>
>
> ## Response to Weaknesses
>
> ### Response to Theoretical contribution in comparison to existing works
>
> Thank you for recommending these papers. We are happy to discuss papers [1,2,3,4] and thank you for the opportunity to clarify our primary theoretical contribution. We first clarify the key theoretical distinction between these papers and our work, and then discuss each referenced paper in more detail.
>
> ### First part:
> **Key theoretical distinction**
>
> Papers [1][2][3][4] all work with related topics and state that their proposed methods satisfy counterfactual fairness under certain conditions. However, none of these papers consider or discuss the identifiability of counterfactuals. These papers **either do not explicitly state their implied stronger assumptions** or **fully ignore the identifiability of counterfactuals** at all and hope their method can somehow manage to learn the correct counterfactuals. Yet, their methods can fail and thus learn predictions that are actually **unfair**. This is because **identification is different from estimation**. Methods that act as **heuristics may return estimates that do not correspond to the true value**.   Therefore, these methods do not have any theoretical guarantee and can lead to **unfair predictions**.
>
> We break down this paragraph and make it more clear point by point below:
>
> (1) **Identification is different from estimation**.
>
> In causal inference, one can think about identifiability as the mathematical condition that permits a causal quantity to be measured from observed data. We kindly refer to papers [5][6], which all discuss in greater depth what identification means in causality. Importantly, identification is different from estimation because methods that act as heuristics may return estimates but they do not correspond to the true value. (see paper [7] where the authors provide several concerns that, if it is not unique, it is possible to have local minima, which leads to **unsafe results**.
>
> (2) **None of these papers considers or discusses the identifiability of counterfactuals**.
>
> (a) Paper [2] claims that they form the counterfactual prediction by drawing from the counterfactual distribution. However, it does **not clarify how they exactly managed to learn this counterfactual distribution**, and neither does the paper prove **why the learned counterfactual distribution by this model should be identifiable**. They just give a **conceptual algorithm**, yet can return an unfair prediction by their predictive model.
>
> (b) Papers [1][3][4] involve the step of learning the latent variables and later training a prediction model using these inferred latent variables. These papers do not have any proper identification guarantees for learned latent variables from data and a given causal graph. However, **Non-identifiability of the latent variables means non-identifiability of the counterfactual queries**. We kindly refer to the paper for that [8].  Again, this leads to predictions that can be actually unfair.
>
> Hence, papers [1,2,3,4] can not ensure that they correctly learn counterfactual fairness with theoretical guarantees. In contrast, our is the only method that does so.
>
> (3)  **Regarding our method**
>
>  We are the first paper in counterfactual fairness literature **not only to propose a tailored model specific to this task but also to provide the necessary assumptions needed for the identifiability of counterfactual and the theory proof specific for our tailored model to ensure it actually achieves real counterfactual fairness** (see **our Lemma 1 and our Lemma 2**).
>
> Lemma 1 is about under which conditions we can have the identifiability of counterfactuals and how our designed model converges to the true distribution of counterfactuals. Further, Lemma 2 is about the fairness level of the prediction being upper-bounded. Thus, we are the first to have theoretical guarantees in correctly learning counterfactual fairness.
>
> (Continue)

---

> ### Author Response · Authors · 2024-11-20
> **Response to Reviewer jvSK**
>
> ### Second part
>
> **Discuss each referred paper separately in more detail**
>
>
> ### 1. Comparison with [1] ( Counterfactual fairness)
>
> [1]: _About Kusner's paper:_
>
> [1] introduced a conceptual algorithm to achieve predictions under counterfactual fairness. The idea is to first infer a set of latent background variables and subsequently train a prediction model using these inferred latent variables and non-descendants of sensitive attributes
>
> What we say “the conceptual algorithm requires knowledge of the ground-truth structural causal model” means that: **Without knowing the ground-truth structural causal model, the paper [1] can not achieve counterfactual fairness.**
>
> Kusner requires the specification of structural equations $F$ for the calculation of counterfactual quantities. This is much more than knowing the domain knowledge about the descendants of the sensitive attributes. This is because the state of any observable variable needs to be fully determined by the background variables and structural equations.
>
> That is to say, **paper [1] can not directly learn from data and the causal graph, but instead needs to know the ground-truth structural causal model**. Our method can learn from data and the causal graph, but needs the assumption to ensure identifiability of counterfactuals.
>
> Additionally, **paper [1] provided only the conceptual algorithm, while only later works proceeded by offering actual instantiation.** Therefore, [1] did not provide any theoretical guarantees for achieving counterfactual fairness in the **prediction models.**

---

> ### Author Response · Authors · 2024-11-20
> **Response to Reviewer jvSK**
>
> ### Comparison with paper [2] (Adjusting Machine Learning Decisions for Equal Opportunity and Counterfactual Fairness)
>
>
> This paper introduces a new fairness criterion called “equal counterfactual opportunity (eco) “ and shows that eco-fairness is equivalent to conditional demographic parity. The paper [2] first trains a classical machine learning predictor to predict the outcome based on all the information (including the sensitive attribution and its descendants). Next, the paper adjusts the decisions made by classical ML to be eco-fair and then to counterfactual fair.  (Some notations are differently used in this paper. In [2], what is referred to as $S(a)$ aligns with $M_a$ in our paper.)
>
> More formally, the paper [2] computes the counterfactual decision by drawing from a mixture distribution following Eq. (2) in paper [2]. The authors propose to sample the counterfactual mediator from the counterfactual distribution. However, there are two clear limitations. First, [2] did **not clarify how they exactly managed to learn this counterfactual distribution** (for example, by what neural models). Second, [2] also did **not prove why the learned counterfactual distribution by this model should be identifiable**.
>
> In contrast, we propose a custom model to learn the counterfactual distribution and give the theory proof of the correctness of the learned counterfactual distribution.
>
>
>
> ### Comparison with paper [3] (Counterfactually fair representation. Advances in Neural Information Processing Systems)
>
> Paper [3] first learns latent variables $U$ and then uses $U$ to generate multiple counterfactual samples, and then generates counterfactual fair representation based on the counterfactual samples. Paper [3] has the same problem as we stated before. [3] did **not discuss the identifiability of counterfactuals**. There is **no** identifiability of the unobservable variable, and there is **no** theoretical guarantee on the generated counterfactual samples and fair representation. This method thus acts as heuristics and return estimates but they do not correspond to the true value.  In other words, the method from [3] may lead to predictions that are unfair.
>
> Besides, [3] makes stricter assumptions than our paper. The paper [3] separates the descendants of the sensitive attribute (which is $M$ in our paper) into $X_1$ and $X_2$, and $X_1$ is the parent of $X_2$. However, we do not need to know the causal relationships of the variables inside the mediator. In other words, [3] requires **more structural knowledge** than our work.
>
>
> ### Comparison with paper [4] (Counterfactual Fairness by Combining Factual and Counterfactual Predictions)
>
> Paper [4] also involves an additional step of learning the latent variables (here, the task involves latent confounders, which is even harder to model and estimate)  and later training a prediction model using these inferred latent variables.
>
> Paper [4] has the same problem as we stated before. [4] does not discuss the identifiability of counterfactuals.  Hence, paper [4] does **not offer a formal statement on how to ensure that it learns the correct latent variables**, and the paper does not have any **proper identification guarantees for learned latent variables**. The estimated latent variable is not guaranteed to be correctly identified and estimated, leading to risks that predictions that are **unfair**.
>
> ### Summary
> In summary, we are the first paper in counterfactual fairness literature to ensure the identifiability of counterfactuals under mild assumptions (other papers have implied stricter assumptions or do not discuss them at all). We thus propose a tailored model specific to this task and provide a theory for our model to ensure it indeed learns the correct counterfactual distribution. Hence, we can formally state that our method achieves the objective of counterfactual fairness and, therefore, is **the first neural method with theoretical guarantees.**

---

> ### Author Response · Authors · 2024-11-20
> **Response to Reviewer jvSK**
>
> **Actions:** We discussed the above papers with greater care and further spelled out more clearly how our paper is different and thus novel. To this end, **we uploaded a revised PDF** where we made the following improvements.  We discuss the importance of identifiability in our related work section and thereby state the limitations of existing methods (see **our revised Section 2.2**). We explain in greater detail why our assumptions are needed and why our assumptions are necessary compared to existing methods (see **our revised Section 2.2 and Appendix C and D**). We further discuss the importance of identifiability in our method section and thereby state why existing baselines make predictions that are unfair and what the added value of our paper is (see **our Appendix C**). We finally added an in-depth comparison to the above works where we explain why our work is important and how it is novel  (see **our new Appendix D**).
>
>
>
> Reference:
>
> [1] Kusner, M. J., Loftus, J., Russell, C., & Silva, R. (2017). Counterfactual fairness. Advances in neural information processing systems, 30.
>
> [2] Wang, Y., Sridhar, D., & Blei, D. (2023). Adjusting Machine Learning Decisions for Equal Opportunity and Counterfactual Fairness. Transactions on Machine Learning Research.
>
> [3] Zuo, Z., Khalili, M., & Zhang, X. (2023). Counterfactually fair representation. Advances in Neural Information Processing Systems, 36, 12124-12140.
>
> [4] Zhou, Z., Liu, T., Bai, R., Gao, J., Kocaoglu, M., & Inouye, D. I. (2024). Counterfactual Fairness by Combining Factual and Counterfactual Predictions. arXiv preprint arXiv:2409.01977.
>
> [5] Pearl, Judea. Causal inference in statistics: An overview. 2009
>
> [6] Peters, Jonas, Dominik Janzing, and Bernhard Schölkopf. Elements of causal inference: foundations and learning algorithms. The MIT Press, 2017.
>
> [7] D’Amour A. On Multi-Cause Causal Inference with Unobserved Confounding: Counterexamples[J]. Impossibility, and Alternatives, 2019.
>
> [8] Melnychuk, Valentyn, Dennis Frauen, and Stefan Feuerriegel. Partial Counterfactual Identification of Continuous Outcomes with a Curvature Sensitivity Model.  Neurips 2023.

---

> ### Author Response · Authors · 2024-11-20
> **Response to Reviewer jvSK**
>
> ### Response to “Causal Model”
>
> Thank you for your suggestion. Yes, we are happy to offer real-world examples to illustrate the role of each variable. In our causal graph, the mediators $M$ are simply all variables that can potentially be influenced by the sensitive attribute. All other variables (except for $A$ and $Y$) are modeled as covariates $X$. For example, consider a job application setting where we want to avoid discrimination by gender. Then $A$ is gender, and $Y$ is the job offer. Mediators $M$ are, for instance, education level or work experience, as both are potentially influenced by gender. In contrast, age is a covariate $X$  because it is not influenced by gender. In sum, it is straightforward to choose the variables in practice (and does not require any domain knowledge or other expertise).
>
> **Action:** We expanded our description of the causal model and linked to our practical considerations of how to choose the variable (see **our Appendix E**).
>
>
> ### Response to “Presentation”
>
> Thank you. We followed your suggestions closely and improved our presentation as result (see the changes in **blue color** in our revised PDF).
>
> 1. Thank you. Yes, the theoretical derivation requires the function to be invertible wrt. to noise ($f_M$ is a strictly increasing (decreasing) continuously-differentiable transformation wrt. $u_M$). We state the assumption for identifiability in Lemma 1, meaning that $f_M$ needs to be the bijective generation mechanism (BGM). We now added this explicitly to the description of our setting (see Sec.3 and Sec 4). We also expanded our discussion of our causal model and why the counterfactual identification assumptions are standard in the literature [5,6,7,8].
>
> 2. In the original Standard Fairness Model, the $X$ is considered an _observed_ confounder. $U$ is considered as a _hidden_ confounder in [1,3,4]. Hence, $X$ is accessible in the observational data, which is consistent with our paper.
>
> Reference:
>
> [1] Kusner, M. J., Loftus, J., Russell, C., & Silva, R. (2017). Counterfactual fairness. Advances in neural information processing systems, 30.
>
> [2] Wang, Y., Sridhar, D., & Blei, D. (2023). Adjusting Machine Learning Decisions for Equal Opportunity and Counterfactual Fairness. Transactions on Machine Learning Research.
>
> [3] Zuo, Z., Khalili, M., & Zhang, X. (2023). Counterfactually fair representation. Advances in Neural Information Processing Systems, 36, 12124-12140.
>
> [4] Zhou, Z., Liu, T., Bai, R., Gao, J., Kocaoglu, M., & Inouye, D. I. (2024). Counterfactual Fairness by Combining Factual and Counterfactual Predictions. arXiv preprint arXiv:2409.01977.
>
>
> [5] Xia K, Pan Y, Bareinboim E. Neural causal models for counterfactual identification and estimation. arXiv 2022
>
> [6] Arash Nasr-Esfahany, Mohammad Alizadeh, and Devavrat Shah. Counterfactual identifiability of
> bijective causal models. In ICML, 2023.
>
> [7] Zhang J, Tian J, Bareinboim E. Partial counterfactual identification from observational and experimental data. ICML 2022
>
>
> [8] Valentyn Melnychuk, Dennis Frauen, and Stefan Feuerriegel. Partial counterfactual identification of continuous outcomes with a curvature sensitivity model. In NeurIPS, 2023.

---

> ### Author Response · Authors · 2024-11-20
> **Response to Reviewer jvSK**
>
> ### Response to “Presentation”
>
> 3. Thank you. We now answer why our method is “non-trivial” and thereby explain the benefits of our method (together with providing an answer to Question 2 from your review).
>
> _What existing methods do:_
>
> Existing methods involve the step of learning the latent variables and later training a prediction model using these inferred latent variables. They often use VAEs to learn the inferred latent variable $U$, we clarify why VAE-based methods for our task can be problematic and why our proposed method is superior – both theoretically and numerically.
>
> _What existing methods are problematic:_
>
> The inferred latent variable should be independent of the sensitive attribute $A$ while representing all other useful information from the observation data. However, there are two main challenges: (1) The latent variable $U$ is **not** identifiable. (2) It is very hard to learn such $U$ to satisfy the above independence requirement, especially for high-dimensional or other more complicated settings. In sum, there are **no** theoretical guarantees for the VAE-based methods. Hence, it is mathematically **unclear** whether they actually learn the correct counterfactual fair predictions. => In other words, existing methods have an objective function that can lead predictions that are unfair.
>
> In fact, there is even rich empirical evidence that VAE-based methods are often **suboptimal** [1,2,3]. VAE-based methods use the estimated variable $U$ in the first step to learn the counterfactual outcome $\mathbb{P}\left(\hat{Y}_{ a'}(\mathbf{U}) \mid X=x, A=a, M=m \right).$ The inferred, non-identifiable latent variable can be correlated with the sensitive attribute which may harm fairness, or it might not fully represent the rest of the information from data and harm prediction performance.
>
> _How our method overcomes the above challenges:_
>
> We address the above challenges by learning the counterfactuals directly. Thus, our method eliminates the need for an abduction-action-prediction process that learns $U$ but, instead, we learn $U$ directly in a single step through our GAN. To this end, we avoid the complexities and potential inaccuracies of inferring and then using latent variables. More formally, we generate counterfactual samples from the learned counterfactual distribution. For each specific input data $(X=x, A=a,M=m)$, we then generate the counterfactual mediator from the distribution $\mathbb{P}(M_{a'} \mid X=x, A=a, M=m)$. This results in overall more accurate and robust predictions.
>
>
>
> ## Answer to “Questions”
>
> 1. Thank you. In Figure 1, we do not assume a direct arrow from $A$ to $Y$. This is because we believe that, in the real world, it is generally not allowed to give direct discrimination in decision-making. For example, a company’s recruitment is – by law – not allowed to make decisions that depend on the gender of an applicant. Similarly, the recidivism of a person should not directly rely on the race of this person. Figure 1 is the causal graph for our model to ensure.
>
> 2. Thank you. We kindly refer to our answer from above regarding your question about “Response to presentation”.
>
> Reference:
>
> [1] Takahashi, Hiroshi and Iwata, Tomoharu and Yamanaka, Yuki and Yamada, Masanori and Yagi, Satoshi. "Variational autoencoder with implicit optimal priors." In AAAI, 2019.
>
> [2] Hoffman, Matthew D., and Johnson, Matthew J. ELBO surgery: yet another way to carve up the variational evidence lower bound. In Workshop in Advances in Approximate Bayesian Inference,  NeurIPS. 2016.
>
> [3] Aaron van den Oord, Oriol Vinyals, koray kavukcuoglu. Neural Discrete Representation Learning. In  In NeurIPS.2017

---

> > ### Comment · Reviewer_jvSK · 2024-11-21
> > **Response to the rebuttal**
> >
> > Thank you for the rebuttal.
> >
> > **Summary**: Overall, I think the authors have addressed my major concerns regarding the clarity of their contributions by providing a more careful discussion in the revised draft. While I find that their theoretical results rely on somewhat restrictive assumptions, such as observed confounders and invertibility, their work could provide useful insights to the Counterfactual Fairness community.
> >
> > I have adjusted my score accordingly.

---

### Official Review · Reviewer_4kNp · 2024-11-04

**Soundness:** 2
**Presentation:** 1
**Contribution:** 1
**Rating:** 3
**Confidence:** 5

**Summary:**

This paper addresses the challenge of achieving counterfactual fairness by proposing a novel approach that learns the counterfactual distribution of the descendants of sensitive attributes using specifically designed neural networks. Fair predictions are enforced through a counterfactual mediator regularization, which aims to minimize the influence of sensitive attributes. Experimental results demonstrate the effectiveness of this architecture.

**Strengths:**

1. The identifiability of counterfactual fairness is an important problem.

2. Experimental results showcase the effectiveness of the proposed architecture.

**Weaknesses:**

1. The paper assumes that the function  $f_M$  is a bijective generation mechanism, which is a strong assumption, particularly in real-world datasets where such conditions are rarely met. While this assumption holds in synthetic datasets, it limits the method’s applicability in fairness contexts that generally involve real-world data. Therefore, the claim that this is the first method for counterfactual fair predictions with theoretical guarantees might be overstated given the challenges with real data. The experiment on real data demonstrates the anticipated outcomes that the proposed GAN architecture can achieve. However, there appears to be a disconnect between the empirical application and the theoretical results; the practical implementation and theoretical guarantees seem to function as separate aspects without a relationship.

2. Lemma 1 seems to be a straightforward extension of results from previous studies (e.g., Lemma B.2 in Nasr-Esfahany et al., 2023, and Corollary 3 in Melnychuk et al., 2023) to GANs. It’s plausible that similar results could be obtained for other neural models with continuously differentiable functions, raising the question of whether the paper’s theoretical contributions are specific to GANs or could be generalized to other architectures.

3. Although the reconstruction loss is employed to ensure the similarity between generated factual mediators and observed factual mediators, there is no explicit assurance regarding the accuracy of generated counterfactuals. This limitation raises concerns about the robustness of counterfactual predictions produced by the model.

4. Lemma 2 establishes an upper bound on the effect of the sensitive attribute on the target variable by focusing on the performance of counterfactual mediator generation and regularization. However, this upper bound does not imply the model’s effectiveness in achieving counterfactual fairness. It also leaves open questions regarding the extent to which the proposed method truly guarantees counterfactual fairness in real-world applications.

**Questions:**

Please refer to the weaknesses above.

---

> ### Author Response · Authors · 2024-11-20
> **Response to Reviewer 4kNp**
>
> Thank you for your review and your helpful comments! We took all your comments at heart and improved our paper as a result. We highlighted all major changes in **blue color** in our **updated PDF**.
>
>
> ## Response to Weaknesses
>
>
> ### Response to W1
>
> Thank you for your comment. We are happy to explain our assumptions and why these are standard in the causal literature [1,2,3,4,5,6,7].
>
>
> (1) It is true that the bijective generation mechanism is required to ensure the identifiability of our method.  The BGM assumption is crucial for the identification of the counterfactuals. It includes many popular identifiable SCMs as special cases, e.g., ANM (Peters et al., 2014)[1], LSNM (Immer et al., 2022)[2], and PNL (Zhang & Hyvarinen,2009)[3]. Thus, this assumption can be seen as **one of the most general assumptions that lead to point identifiability**.
>
>
> (2) We agree that it is more challenging for real datasets to meet the assumption. It is hard to argue whether real-world datasets usually satisfy the BGM assumption. Rather, this assumption provides a guideline for which datasets it is in principle possible to provide answers to the counterfactual questions and for which not. Hence, we offer new theoretical insights regarding which scenarios can counterfactual fairness be achieved in the first place (and which not). As discovered by [8], the relaxation of the BGM assumption not only immediately leads to point non-identifiability but also to non-informative partial identification bounds. Still, it can be intuitively re-formulated ([4]) as follows: In $f_M$, the sensitive attribute, $A$, is assumed to interact only with the observed covariates, $X$, and not with the exogenous noise, $U_M$. Many real-world data-generation mechanisms / phenomena, if studied closely, can be said to satisfy this assumption, e.g., simulators in physics and medicine but also neuroscience and behavioral processes.
>
> (3) **Counterfactual identification is a challenging and very difficult task, and current literature on this direction requires strong assumptions to fulfill it** [4,5,6,7]. Hence, our paper is consistent with the assumptions in [4,5,6,7].
>
> (4) We can have a theoretical guarantee on the datasets that satisfy this assumption. There is no disconnection between the empirical application and the theoretical results of these datasets.
>
> We would like to stress that **other methods do not even have any theoretical guarantees on the identifiability** of their methods or provide **no guarantees of the correctness of counterfactual fairness achieved** on either synthetic datasets or real-world datasets.  We believe having some assumptions to make our method **with theoretical guarantees is better** than methods with no correctness guarantee at all. => In other words, existing methods **either do not explicitly state their implied stronger assumptions** or **lack the identification of counterfactuals** (i.e., the ability of the method to determine the true value).  These methods all act as **heuristics** that may return estimates but these estimates do not correspond to the true value. Therefore, these methods do not have any theoretical guarantee and can lead to **unfair predictions**.
>
> **Action:** We have added explanation in greater detail why our assumptions are standard in the counterfactual identification literature and why it is **important and necessary for counterfactual fair prediction** (see our revised **Section 2.2 and Appendix C and D**)
>
> Reference:
>
> [1] J. Peters, J. Mooij, D. Janzing, and B. Scholkopf. Causal discovery with continuous additive noise models. Journal of Machine Learning Research.
>
> [2] Alexander Immer, Christoph Schultheiss, Julia E Vogt, Bernhard Scholkopf, Peter Buhlmann, and Alexander Marx. On the identifiability and estimation of causal location scale noise models. arXiv preprint.
>
> [3] K. Zhang and A. Hyvarinen. Distinguishing causes from effects using nonlinear acyclic causal models. In Proc. Workshop on Causality: Objectives and Assessment at NIPS 2008
>
> [4] Valentyn Melnychuk, Dennis Frauen, and Stefan Feuerriegel. Partial counterfactual identification of continuous outcomes with a curvature sensitivity model. In NeurIPS, 2023.
>
> [5] Xia K, Pan Y, Bareinboim E. Neural causal models for counterfactual identification and estimation. arXiv 2022.
>
> [6] Arash Nasr-Esfahany, Mohammad Alizadeh, and Devavrat Shah. Counterfactual identifiability of
> bijective causal models. In ICML, 2023.
>
> [7] Zhang J, Tian J, Bareinboim E. Partial counterfactual identification from observational and experimental data. ICML 2022

---

> ### Author Response · Authors · 2024-11-20
> **Response to Reviewer 4kNp**
>
> ### Response to W2
>
> Thank you.  Lemma 1 is not a straightforward extension of results from previous studies. This is because of three reasons:
>
> (1) We first propose a **tailored GANs architecture which allows for an end-to-end training for this task**. Note that this is not an off-the-shelf GAN but one that is closely designed to achieve our theoretical guarantees. To address the problems of baseline methods that infer with latent variables, we designed our **custom** GAN architecture in a way so that we avoid the complexities and potential inaccuracies of inferring and then using latent variables. Instead, we use the adversarial training process to encourage the generator to learn the counterfactual distribution of the mediator.
>
> (2)  Previous studies (Lemma B.2 in Nasr-Esfahany et al., 2023, and Corollary 3 in Melnychuk et al., 2023) aim to discuss counterfactual identifiability. However, **our Lemma 1.2 is a new theoretical result regarding counterfactual identifiability with our tailored GANs**. It states that our generator consistently estimates the counterfactual distribution of the mediator $\mathbb{P}(M_{a'} \mid X=x, A=a, M=m)$. **This provides a theoretical justification behind the design of our generator for the counterfactual mediators**.
>
> (3) Additionally,  Lemma 1 serves as the basis for the Lemma 2. The entire Lemma 2 is our contribution (and not from existing work). Together, both Lemma 1 and Lemma 2 effectively prove that we can enforce the predictor to be counterfactually fair.

---

> ### Author Response · Authors · 2024-11-20
> **Response to Reviewer 4kNp**
>
> ### Response to W3
>
> Thank you for raising this important point. You are right in pointing out that, during training, we cannot directly learn the counterfactual mediators in a supervised way as they are unobservable. Instead, we can only leverage the reconstruction loss on the factual mediators. **The reason why we can still learn the correct counterfactual mediators is due to the adversarial training process of the generator**. By training the discriminator to differentiate between factual and generated counterfactual mediators, the generator is guided to learn the correct counterfactual distribution. It could be seen as a form of **teacher forcing**.
>
> We are also happy to explain more about why our method does **not** simply reproduce the **factual** mediators but actually learns the **counterfactual** mediators even though it is not observed. There are three important arguments for this:
>
> (1) **Intuitively**: The input of the discriminator $D$ contains the counterfactual and the factual and the _order of them is intentionally randomized_. Suppose, hypothetically, that the factual always comes in the first place, then it is easy to distinguish. However, in the design of our framework, this is **not** the case. In our framework, the data are intentionally shuffled, so that factual and counterfactual positions are random.
>
> (2) **Technical reason**: If the generator $G$ would just copy the factual of the mediator $M$ and output a trivial solution, then the same would happen as you mentioned: it would be super hard for the discriminator $D$ to distinguish both, implying that the loss of $D$ would be super large. We would thus observe mode collapse during training. Yet we did **not** observe.
>
>
> (3) **Theoretical reason**: We provide theoretical proof in our Lemma 1. Therein, **we show theoretically that our generator consistently estimates the counterfactual distribution of the mediator** $\mathbb{P}(M_{a'} \mid X=x, A=a, M=m)$. For each specific input data $(X=x, A=a,M=m)$, we then generate the counterfactual mediator from the distribution $\mathbb{P}(M_{a'} \mid X=x, A=a, M=m)$. Hence, **we offer theoretical proof that we learn the counterfactual values correctly**.
>
>
> Therefore, we performed **additional experiments** (see **Appendix K.2**). We observed that (1) The factual mediator and the generated counterfactual mediator are highly dissimilar. (2) The ground-truth counterfactual mediator and our generated counterfactual mediator are highly similar. This confirms that the mediator is not simply copying the factual values but learning the distribution of counterfactual values.
>
>
> **Action:** We have added the above explanation and experiments in **our new Section 5.2.2 and Appendix K.2** in our revised paper.

---

> ### Author Response · Authors · 2024-11-20
> **Response to Reviewer 4kNp**
>
> **Response to W4**
>
> Thank you for this comment. Lemma 2 states that the influence of the sensitive attribute on the target variable is upper-bounded by (i) the estimation of counterfactual mediators (first term) and (ii) the counterfactual mediator regularization (second term).
>
> We are happy to explain both in greater detail: (i) The first term does not depend on $h$, and, given Lemma 1, reduces to zero as there exists a generator in $\mathcal{G}$, which consistently estimates counterfactuals. By reducing (ii) the second term $\mathcal{R}_\mathrm{{cm}}$ for all the generators through minimizing our training loss, we can effectively enforce the predictor to be more counterfactual fair.
>
> The upper bound is valid for every counterfactual generator that minimizes our adversarial objective and not just for the arg-supremum generator. Furthermore, as there exists a generator that consistently estimates counterfactuals, the first term of the bound decreases to zero with a sufficient amount of data. **The upper bound indeed guarantees the model’s effectiveness in achieving counterfactual fairness.** We also have shown the proposed method truly guarantees counterfactual fairness in our experiments.

---

> ### Comment · Reviewer_4kNp · 2024-11-22
> **A Summary of More Specific Concerns**
>
> Thank you for the clarification.
>
> While I agree that BGM assumption is one of the most general assumptions leading to point identifiability, I believe the claim that the proposed method is the first for counterfactual fair predictions with theoretical guarantees is imprecise. I will elaborate on my concern more specifically in the following.
>
> **According to the first statement of Lemma 1 (directly from Lemma B.2 in Nasr-Esfahany et al., 2023, and Corollary 3 in Melnychuk et al., 2023), counterfactual  M  can be identifiable under certain assumptions. Once these assumptions are satisfied, it seems that any generative model could, in principle, be employed to generate counterfactuals,** including those that require learning latent variables first, such as VAEs. After all, identifiable counterfactuals inherently imply the identifiability of latent variables (as the authors noted, “Non-identifiability of the latent variables means non-identifiability of the counterfactual queries”).
>
> My main point is this: why is it necessary to use GAN specifically, apart from design considerations? **From my understanding, GAN is just one of many possible models that could be utilized for estimation in this setting. Therefore, the critical contribution should focus on how the proposed GAN ensures the statistical efficiency and consistency of the estimated counterfactuals, addressing issues such as estimation bias and variance and their impact on prediction effectiveness.** While Lemma 1.2 touches on this by requiring that the GAN generator be a continuously differentiable function with respect to  M , this condition does not appear to be particularly restrictive. I am curious whether a similar condition might also render VAE applicable in this context, particularly considering the authors' claim that abduction methods like VAE cannot achieve this. Moreover, I do not think the upper bound presented in Lemma 2 implies the effectiveness of the model considering the quality of generated counterfactuals.
>
> **In summary of my concerns,**
>
> 1. **Regarding the contributions on estimation,** related to statistical efficiency and consistency of the generated counterfactuals (e.g., bias, variance) and the effectiveness of predictions (which depends on the quality of the estimated counterfactuals), I believe these aspects are not fully explored in the paper. This lack of discussion is the basis for my concerns related to Weakness 3 on quality of generated counterfactual and Weakness 4 on the effectiveness of the prediction.
>
> 2. **Moreover, with respect to the identifiability of counterfactuals** in the context of counterfactual fair predictions, this contribution is a relatively straightforward extension of Lemma B.2 in Nasr-Esfahany et al., 2023, and Corollary 3 in Melnychuk et al., 2023.
>
> 3. **The authors claim that their method is the first to offer counterfactual fair predictions with theoretical guarantees. It seems such theoretical guarantee focus on the estimation aspect --- specifically, “the generator of GAN is a continuously differentiable function with respect to  M ”. However, such contribution is relatively limited. The estimation is not thoroughly discussed as I mentioned above in 1**, and the limitations of other generative models with respect to this condition, such as VAEs (which seems to be part of the motivation of the work) and the prediction of effectiveness remain unclear.
>
> Given the above-mentioned points, which are either insufficiently addressed or not clearly discussed, the claim that this method is the first to offer counterfactual fair predictions with theoretical guarantees seems overstated or imprecise.

---

> ### Author Response · Authors · 2024-11-22
> **Response to Reviewer 4kNp (Response to: A Summary of More Specific Concerns)**
>
> Thank you for your constructive feedback and your helpful comments.
>
> ###  Response to " it seems that any generative model could, in principle, be employed to generate counterfactuals"
>
>
> Thank you for the questions. We are happy to clarify that our results only hold for GANs, specifically, and it comes with specific advantages.
>
>
>
> * It is true that, any model that for example models both conditional CDFs and conditional quantiles can be used as an estimator of the expected counterfactuals (according to Eq. 14), e.g., conditional normalizing flows. Yet we proved **in the second part of the Lemma**, that **the generator of our GAN estimates the Eq. 14 end-to-end, without explicitly modelling the conditional CDFs and quantiles.** This is a **non-trivial** result, because:
>
> (1) We needed to show that **our GAN minimizes JSD wrt. potential outcomes distributions**;
>
> (2) We **used the property that our GAN is continuously differentiable and deterministic to prove that the potential outcome distribution is actually generated in a way that matches the ground-truth counterfactual distributions**.  This **end-to-end estimation procedure provides our novelty**.
>
> * To the best of our understanding, **an approach with VAE does not necessarily yield consistent estimators of the conditional CDFs and quantiles**. It is easy to see that, by design, **general VAEs yield random latent variables given the input, yet in the case of the BGMs, they have to be deterministic. Therefore, VAEs are misspecified by design for our task**.
>
> In summary, the GAN is **not** just one of many possible models used for estimation in our setting. It comes with **specific advantages** (it gives an **end-to-end estimation and learns the true counterfactual distributions**). Thus, while the study of efficient estimation is indeed an interesting direction for future research, it is not the topic of our paper.
>
>
>
> ###  Response to "concerns":
>
> Thank you.
>
> 1. Regarding statistical efficiency and consistency, we have consistency due to the universal approximation theorem with neural networks (and the BGM solution is continuously differentiable), so the learning target is a well-behaved function. Regarding efficiency, we don’t claim it and it would be an interesting research direction (actually, there isn’t really a practical method, only theory [1] (Conservative Inference for Counterfactuals)).
>
> 2. We would kindly clarify that **it is not that straightforward**, because:  (1) we needed to show that our GAN minimizes JSD wrt. potential outcomes distributions; (2) we used the property that our GAN is continuously-differentiable and deterministic to prove that the potential outcome distribution is actually generated in a way that matches the ground-truth counterfactual distributions.
>
> 3. Our novelty stems from building an **end-to-end model to estimate the BGM solutions**. To the best of our knowledge, no other method was proposed to estimate the BGM solution in such a manner. Ours is the first work in the counterfactual fairness literature to rigorously state the assumptions needed for the identifiability of counterfactuals in our theory and then derive identifiability results for our model. We first propose a **tailored model specific to this task** and further provide **specific theoretical proof** to show that **our model learns the correct counterfactual distribution** and therefore the counterfactual fair predictions.
>
> Reference:
>
> [1] Balakrishnan, Sivaraman, Edward Kennedy, and Larry Wasserman. "Conservative inference for counterfactuals." arXiv preprint (2023).

---

> ### Comment · Reviewer_4kNp · 2024-11-23
> **Issues:**
>
> Thank you for the response.
>
> Before addressing the authors’ reply, I would like to highlight a significant issue. It appears that the authors do not clearly distinguish between “identifiability” and “estimation”, and at times, they conflate the two. This confusion permeates the paper and undermines the clarity of the claims.
>
> **The counterfactual identifiability result is established in the first part of Lemma 1, which is directly derived from Lemma B.2 in Nasr-Esfahany et al., 2023, and Corollary 3 in Melnychuk et al., 2023.** The authors acknowledge that their new contribution lies in the second part of Lemma 1, which pertains to estimation—specifically, estimating the identifiable counterfactuals. (It is worth noting that estimation tasks inherently rely on identifiable counterfactuals in causality; if the counterfactuals are not identifiable, there is no need to discuss the estimation performance.)
>
> **Therefore, it is misleading and overstated for the authors to claim a contribution to identifiability in various parts of the paper, such as Lines 111-115 and the introduction, the motivation part.** The terminology and expressions used regarding "estimation" and "identifiability" are also inconsistent and confusing. For instance:
> - The phrase “estimated incorrectly” is frequently used when referring to other works where counterfactuals are unidentifiable. However, this is unfair and imprecise. When counterfactuals are unidentifiable, the estimation is surely incorrect—it is not a fault of the estimation method itself.
> - Similarly, when counterfactuals are identifiable (i.e., the conditions in Lemma 1.1 are satisfied), the meaning of “incorrectly estimated” becomes unclear. How is the quality of estimation being evaluated in such cases? **Different models (e.g., GANs, VAEs) can be used to perform estimation when identifiability is established, and the distinction between these models lies in estimation performance, not identifiability.**
> - Phrases such as “correctly identified” are also used carelessly. Identifiability is a theoretical property; it cannot be “correct” or “incorrect” in the way estimation results can be.
>
> This conflation of identifiability and estimation leads to casual and imprecise language throughout the paper. If the authors aim to present theoretical results, the rigorous consistent terminology is required.
>
>
> ### Key Concerns:
>
> 1. Focus on Estimation but Lack of Exploration of Statistical Efficiency/Consistency.
>
> As I previously noted, the paper is focused on counterfactual estimation, yet it fails to fully explore statistical efficiency and consistency of the proposed estimation. Deferring these critical discussions feels like sidestepping an essential issue, especially when the author aims to claim the superiority of the GAN with respect to other generative models.
>
> The only exploration of consistency is in Lemma 1.2, which requires the GAN generator to be a continuously differentiable function with respect to  M . While I acknowledge this contribution, I do not believe this requirement represents a significant limitation for other generative models. In the authors’ new response, they claim that GANs work here instead of VAEs because of their deterministic nature. However, this distinction seems to be a design choice rather than an inherent advantage. After all, what both the GAN generator and the VAE aim to estimate is the distribution of the potential mediator for counterfactuals, rather than specific point estimates.
>
> As previously discussed, identifiable counterfactuals imply identifiable latent variables. The explanation provided by the authors regarding this point remains unconvincing, and a clear and detailed discussion is needed. Additionally, the reason previous works based on VAEs were unable to establish identifiability may be not primarily due to the VAE structure itself but rather due to the lack of some other assumptions, such as the BGM assumption on the data. A discussion is necessary here.
>
> 2. Claim on Identifiability Contribution is Misleading.
>
> The authors’ repeated claim that their work contributes to identifiability is critically misleading, particularly in the introduction and motivation sections. The identifiability results they present are direct extensions of existing works, as acknowledged in the derivations of Lemma 1.1. The true novelty lies in the estimation procedure, yet the authors overstate their contribution to identifiability.
>
> 3. Theoretical Guarantees on Prediction Effectiveness (Lemma 2).
>
> The authors did not provide additional explanation or justification for the claimed theoretical guarantees on prediction effectiveness in Lemma 2.

---

> ### Comment · Reviewer_4kNp · 2024-11-23
> **Conclusion:**
>
> ### Conclusion:
>
> Given the above issues—particularly the conflation of identifiability and estimation, the lack of exploration into statistical efficiency/consistency, and the misleading claims about identifiability—I find the paper’s presentation problematic. The introduction and motivation sections, in particular, overstate contributions and could mislead readers. Furthermore, the theoretical guarantees on prediction effectiveness (Lemma 2) are inadequately discussed.
>
> For these reasons, I will keep my score.

---

> ### Author Response · Authors · 2024-11-25
> **Response to Reviewer 4kNp**
>
> Thank you for the detailed feedback. We appreciate your review and carefully revise the paper to address the concerns raised.
>
> We acknowledge the importance of maintaining a clear distinction between identifiability and estimation. We agree that some of the language in the manuscript may have been imprecise, leading to potential conflation. **We thus revised the statement in the introduction, related work, and theoretical results parts**, to ensure consistent and rigorous use of terminology. We agree that Lemma 1.1 (identification) may be an implication/consequence of the literature. **Thus we have downgraded the Lemma 1.1 to a remark in our revised paper**.
>
> Our primary contribution lies in Lemma 1.2 about the estimation procedure for identifiable counterfactuals. We have revised the manuscript to ensure that the claims in the introduction and motivation sections do not overstate our contribution to identifiability and instead highlight the novelty in our estimation approach.
>
>
> Still, **none of the previous work in the counterfactual fairness prediction has thought about or discussed identifiability in the first place**. Therefore, we hope that **this still makes a contribution to the field of counterfactual fairness (and addresses a limitation of existing fairness methods that have overlooked this and where the question has not yet been studied in this literature / for fairness practitioners.**
>
>
> We value your constructive review. Thank you again for your detailed and insightful comments.

---

> ### Comment · Reviewer_4kNp · 2024-11-27
>
> Thanks.
>
> First, the overstatements remain pervasive throughout the paper, which could significantly mislead readers who are not familiar with this area, especially counterfacutal identification, estimation, and counterfacutal fairness.
>
> Second, I'd like to emphasize that I do not find the estimation process particularly novel. Many previous works have already utilized GANs for counterfactual fairness [1]. Besides, the authors haven't yet addressed the superiority of GAN over other generative models, like VAE, in estimation.
>
> Third, it is well known that counterfactuals are inherently non-identifiable. For this reason, many prior studies on counterfactual fairness have concentrated on bounding counterfactual queries rather than determining their exact values [2][3][4]. Thus, this work is not the first related to this issue. The closely related works [2][3], which are highly relevant to this study, are not even cited in the paper.
>
> Fourth, the authors have not addressed concerns regarding the quality of the estimation (i.e., the quality of the generated counterfactuals) in Lemma 1 and the prediction effectiveness in Lemma 2.
>
> Lastly, it remains unclear how Lemmas 1 and 2 contribute to the experiments. What is the specific motivation behind this work, and how do these lemmas translate into practical significance?
>
> Given these critical issues, I am keeping my score.
>
> [1] Depeng Xu, Yongkai Wu, Shuhan Yuan, Lu Zhang, and Xintao Wu. Achieving causal fairness through generative adversarial networks. In IJCAI, 2019.
>
> [2] Wu, Yongkai, Lu Zhang, and Xintao Wu. "Counterfactual fairness: Unidentification, bound and algorithm." Proceedings of the twenty-eighth international joint conference on Artificial Intelligence. 2019.
>
> [3] Wu, Yongkai, et al. "Pc-fairness: A unified framework for measuring causality-based fairness." Advances in neural information processing systems 32 (2019).
>
> [4] Zhang, Junzhe, Jin Tian, and Elias Bareinboim. "Partial counterfactual identification from observational and experimental data." International conference on machine learning. PMLR, 2022.

---

> ### Author Response · Authors · 2024-11-27
> **Response to Reviewer 4kNp**
>
> Thank you for the response. We appreciate your detailed feedback.
>
> 1. We thank the reviewer for pointing out the importance of clearly stating the contribution of the paper. We have revised the introduction and related work in our paper to maintain a clear distinction between identifiability and estimation. We ensure the consistent and rigorous use of terminology and make it more clear about our contribution. Our main contribution in this paper is **discussing the identifiability of counterfactuals explicitly in the field of counterfactual fairness and providing a tailored method to link the gap between identifiability and estimation in this fairness literature**.
>
> 2. Thank you. We would like to kindly clarify that we did not state our novelty is using GANs, the GANs are just a tool, instead, our novelty is in our framework can ensure fairness with the theoretical guarantee by **consistently estimating counterfactual distribution (Lemma 1.2)** under the explicitly discussed identifiability of counterfactual assumption and **upper bound fulfilled by our model (Lemma 2)**.  In this way, we thus effectively **enforce the predictor to learn counterfactual fair predictions**. We think GANs are suitable for this task (after our tailored modification), thus we **provided the specific proof** of the consistent estimation **using our tailored model**.  As for paper [1], it is vastly different from our method in many aspects, and we already provided a detailed comparison in Appendix D.6. We had Appendix D2 and Appendix D3 discussing why our method is superior to VAE baselines on this task. We kindly clarify that our novelty is **not about what kind of generative model is used, but about providing a theoretical guarantee on our end-to-end training procedure to ensure the counterfactual fairness prediction, which none of the previous work has done**.
>
> 3. Thank you. We agree that counterfactuals are inherently non-identifiable. There are many papers that focus on partial identifiability and give bounds instead, however, this is beyond our work. We explicitly state the identifiability assumption required in our work, thus we can later base on it and ensure the next estimation step. It could be possible for further work to provide other theoretical guarantees under other counterfactual identifiability assumptions (e.g.,  partial identifiability).
>
> 4. We gave the empirical evidence of the quality of the counterfactual estimation (Lemma 1) in Appendix K.2, which shows that the generated counterfactual by our model is similar to the ground-truth counterfactual. We also proved the prediction effectiveness (Lemma 2) in our benchmarking experiments (e.g., Section 5.2.1).
>
> 5. Lemma 1 and 2 provide the theoretical guarantees for our method. They **provide theoretical proof of why our method can be empirically better than other methods**. Our method has shown superior performance on this task. **Lemma 1 and 2 are consistent with our experiment results** .  **They together bring a significant step forward toward a safe and reliable use of counterfactual fairness in practice**.
>
>
> We value your constructive review. Thank you again for your detailed and insightful comments.

---

> ### Author Response · Authors · 2024-12-02
> **Response to Reviewer 4kNp**
>
> Dear Reviewer,
>
> Thank you for your thoughtful feedback and for taking the time to review our submission.
>
> We have incorporated all action points into our revised paper. We hope we have addressed all the concerns you raised in your review, and have made it clear for the contribution of this paper in the counterfactual fairness prediction field.
>
> Thank you again for your constructive comments, which have helped improve the quality of our submission.

---

### Official Review · Reviewer_d2d9 · 2024-11-04

**Soundness:** 1
**Presentation:** 3
**Contribution:** 2
**Rating:** 5
**Confidence:** 3

**Summary:**

The authors introduce a framework called the Generative Counterfactual Fairness Network (GCFN), which is designed to produce predictions that are counterfactually fair. Their approach involves a two-step process using Generative Adversarial Networks (GANs) to learn the distribution of counterfactuals of sensitive attribute mediators. This method is unique in that it directly models counterfactual mediators without requiring latent variable inference, a step that previous methods used but lacked guarantees of fairness.

**Strengths:**

1. Counterfactual fairness is crucial since it directly addresses biases related to sensitive attributes. Counterfactual fairness ensures fairness at an individual level.

2. The paper is well-written and easy to follow.
3. The proposed method is evaluated through different settings on multiple datasets.

**Weaknesses:**

I have multiple concerns regarding the theory, objective and experiments. Please see questions below.

**Questions:**

1. The objective requires training multiple GANs, which may not be feasible for lots of real-world tasks.

2. Lemma 1: The cited Corollary 3 (Melnychuk et al., 2023) is valid only for real-valued random variables. If the mediator is assumed to be scalar, this assumption should be explicitly stated. Moreover, assuming real-valued mediators is restrictive.

3. The performance heavily depends on the quality of the generated counterfactual mediator. How is this ensured in your objective? While the objective includes both adversarial loss and reconstruction loss, it is unclear how this setup could effectively control the quality of the generated counterfactual mediators. For example, if the generated counterfactual mediator $\tilde{M}_{a’}$ is distributionally aligned with the factual data $M$ given $A=a'$, how can a discriminator distinguish between them using your objective? If the discriminator is unable to differentiate between them, how can we assert that the generated mediators are counterfactual rather than factual samples conditioned on an alternate sensitive attribute value? In such a case, the method would likely address group fairness rather than counterfactual fairness.

4. Why not just use some generative model to generate counterfactuals and train a classifier on top of it? Is generating the mediator M easier or more accurate compared to directly generating the counterfactual $X$?

5. In Appendix D, why does equation 18 hold? What does $\\{M, G(X,0,M)_1\\}$ mean? And why $\\{M, G(X,0,M)_1\\}=\tilde{G}$?

---

> ### Author Response · Authors · 2024-11-19
> **Response to Reviewer d2d9**
>
> Thank you for your review and your helpful comments! We took all your comments at heart and improved our paper as a result. We highlighted all major changes in **blue color** in our **updated PDF**.
>
> ## Answer to “Questions”
>
> ### Answer to Q1:
>
> Thank you for asking this important question. We would like to clarify that it is **not necessary** to use multiple GANs for empirical performance. On the contrary, we show that even a single GAN achieves state-of-the-art performance (see our new Section 5.2.2 and our new Appendix K.3 ). We summarize the results in the following:
>
> We initially used the approach based on several GANs to ensure that the assumptions of our theoretical foundation were met. However, this was merely done for **theoretical reasons**, but **not** better performance. More specifically, we train several GANs to ensure the **worst-case** counterfactual fairness to be aligned with our theory. This gives the upper bound of the extent to which counterfactual fairness is fulfilled in the prediction model, which is needed for our **theoretical guarantees**.
>
> In fact, we would like to highlight that, in practice, the scalability of our proposed method can be simplified using a single GAN. The results are shown in **Sec.5.2.2 and Appendix K.3**  in our revised PDF. We can see that **our method based on a single GAN still gives good results and outperforms all baselines**. Therefore, we recommend using a single GAN for empirical use and multiple for theoretical guarantees.
>
> **Action**: We added results with a single GAN to our main paper (see **our new Subsection 5.2.2 and our Appendix K.3**).
>
>
> ### Answer to Q2:
> Yes, you are right. Thank you for your suggestion. We have made this assumption explicitly stated in our paper. It is very challenging to prove the counterfactual identifiability in high-dimensionality. We have put it as the future works of our paper.
>
>
> ### Answer to Q3:
>
> Thank you for asking this important question.
> (1). In our reconstruction loss: we have the ground truth factual mediator, and we use this loss to ensure that our generated factual mediator is similar to the observed factual mediator. (2). In our adversarial loss:  the generator seeks to generate counterfactual mediators in a way that minimizes the probability that the discriminator can differentiate between factual mediators and counterfactual mediators, while the discriminator seeks to maximize the probability of correctly identifying the factual mediator. We replace the generated factual mediator with the observed factual mediator before passing it as input to the discriminator. Hence, it could be seen as a form of **teacher forcing**.
>
> Additionally, we are happy to explain more about why our method does **not** align with (simply reproducing) the **factual** mediators,  but actually learns the **counterfactual** mediators. There are three important arguments for this:
>
> (1) **Intuitively**: The input of the discriminator $D$ contains the counterfactual and the factual and the _order of them is intentionally randomized_. Suppose, hypothetically, that the factual always comes in the first place, then it is easy to distinguish. However, in the design of our framework, this is **not** the case. In our framework, the data are intentionally shuffled, so that factual and counterfactual positions are random.
>
> (2) **Technical reason**:  If the generator $G$ would just copy the factual of the mediator $M$ and output a trivial solution, then the same would happen as you mentioned: it would be super hard for the discriminator $D$ to distinguish both, implying that the loss of $D$ would be super large. We would thus observe mode collapse during training. Yet we did **not** observe.
>
> (3) **Theoretical reason**: We provide theoretical proof in our Lemma 1. Therein, **we show theoretically that our generator consistently estimates the counterfactual distribution of the mediator** $\mathbb{P}(M_{a'} \mid X=x, A=a, M=m)$. For each specific input data $(X=x, A=a,M=m)$, we then generate the counterfactual mediator from the distribution $\mathbb{P}(M_{a'} \mid X=x, A=a, M=m)$. Hence, **we offer theoretical proof that we learn counterfactual values correctly**.
>
> Therefore, we performed **additional experiments** (see our **Appendix K.2**). Therein, we observe that: (1) The factual mediator and the generated counterfactual mediator are highly dissimilar. (2) The ground-truth counterfactual mediator and our generated counterfactual mediator are highly similar.
>
> **Action:** We have added the above explanation and experiments in **our new Section 5.2.2 and Appendix K.2** in our revised paper.

---

> ### Author Response · Authors · 2024-11-19
> **Response to Reviewer d2d9**
>
> ### Answer to Q4:
>
> Thank you for the question.
>
> (1) Yes, it is possible to use other generative models to generate counterfactuals and train a classifier on top of it. It is basically what some of our baselines do [1,2,3]. For example, they use VAE to learn latent variables in the first step, and then use the estimated latent variable to generate the counterfactual outcome.
>
> However, the latent variables in the methods were previously shown to be **non-identifiable**, and non-identifiability of the latent variables means **non-identifiability of the counterfactual queries**. Therefore, there are **no** theoretical guarantees for these methods. Hence, it is mathematically **unclear** whether they actually learn the correct counterfactuals. Rather, these methods learn predictions that may be **unfair**. In contrast, our method is tailored for this task and with theoretical guarantees due to our custom GANs.
>
> (2) We suppose the reviewer is asking if generating the mediators $M$ is easier or more accurate compared to directly generating all counterfactuals together (including the confounder $X$. The answer is yes.
>
> Distinguishing between covariate $X$ and mediator $M$ can make interpretation easier as both have different roles (e.g., $X$ is not affected by the sensitive attribute but $M$ is).
>
> We can generate counterfactual fair predictions more accurately as this helps to remove noise. In simple words, one could simply locate the covariates as part of the mediators, but this is the number of variables that are generated by the GAN and thus leads to more noisy predictions in practice. For instance, using a 100-dimensional feature vector as $M$ can be more complex and difficult for generating its counterfactual than initially partitioning these features into covariates $X$ and mediators $M$ based on domain understanding. The reason is that such a division can significantly reduce the dimensionality (e.g., to just 10 for $M$), thus simplifying the process of generating accurate counterfactuals in the next step and thus improving outcomes. Therefore, generating the mediators $M$ is easier or more accurate compared to directly generating all counterfactuals together.
>
> Reference
>
> [1] Hyemi Kim, Seungjae Shin, JoonHo Jang, Kyungwoo Song, Weonyoung Joo, Wanmo Kang, and Il.Chul Moon. Counterfactual fairness with disentangled causal effect variational autoencoder. In AAAI, 2021.
>
> [2] Pfohl, Stephen R., et al. Counterfactual reasoning for fair clinical risk prediction. Machine Learning for Healthcare Conference. PMLR, 2019.
>
> [3] Grari, Vincent, Sylvain Lamprier, and Marcin Detyniecki. Adversarial learning for counterfactual fairness. Machine Learning 112.3 (2023)

---

> ### Author Response · Authors · 2024-11-19
> **Response to Reviewer d2d9**
>
> ### Answer to Q5:
>
>
> For derivation from Equation (17) to Equation (18):
>
> $\mathbb{E}\_{X \sim \mathbb{P}(X)} \Big[\mathbb{E}_\{{M \sim \mathbb{P}(M \mid X, A = 0)}}\left[\log\big( {D}(X, \tilde{G}\left(X,0,M \right))\_{0}\big) \right ] \ \pi\_0(X)  +  \mathbb{E}\_{({M \sim \mathbb{P}(M \mid X, A = 1)}}\left[\log \big( {D}(X, \tilde{G}\left(X,1,M \right))\_{1}\big) \right] \ \pi\_1(X) \Big]$ (17)
>
> $= \mathbb{E}\_{X \sim \mathbb{P}(X)} \Big[ \mathbb{E}\_{{M\sim \mathbb{P}(M \mid X, A = 0)}}\left[\log \big( {D}(X, \lbrace M, {G}\left(X,0,M \right)\_1\rbrace)\_{0} \big) \right ] \ \pi\_0(X) +  \mathbb{E}\_{{M \sim \mathbb{P}(M \mid X, A = 1)}}\left[\log \big(1 - {D}(X, \{\lbrace {G}\left(X,1,M \right)\_0, M \rbrace\})\_{0} \big) \right ] \ \pi_1(X) \Big]$ (18)
>
> (1) Answer to What does $\lbrace M, {G}\left(X,0,M \right)\_1\rbrace)$ mean:
>
> The $\lbrace M, {G}\left(X,0,M \right)\_1\rbrace$ in Equation(17) means $\tilde{G}(X,0,M)$.
>
> In Equation (4) in our paper,  we give the definition of $\tilde{G} \left(X, A, M\right)$. We modify the output of $G$ before passing it as input to the discriminator $D$: We replace the generated factual mediator $\hat{M}_A$ with the observed factual mediator $M$. We have the new, combined output denoted by $\tilde{G} \left(X, A, M\right)$, which is used as the input of the discriminator $D$ in the next step.
>
> $\tilde{G}(X,0,M)$ denotes $\tilde{G} \left(X, A, M\right)$ when $A=0$. We can write it in the form $\lbrace M, {G}\left(X,0,M \right)\_1\rbrace$. We show why this is the case below in (2).
>
> (2) Answer to why $\lbrace M, {G}\left(X,0,M \right)\_1\rbrace = \tilde{G}$.
>
> In Equation (4) in our paper, we have
> $\tilde{G}(X,A,M)_a=\begin{cases} M, & \text{if } A=a \\\\ G(X,A,M)_a, & \text{if } A=a' \end{cases}$
>
> For the Equation (17) first term, the mediator $ M $ is drawn from $ \mathbb{P}(M \mid X, A = 0) $. For $A=0$,
> $\tilde{G}(X,0,M)_a=\begin{cases} M, & \text{if } A=a \\\\ G(X,A,M)_a, & \text{if } A=a' \end{cases}$
>
> Therefore, we can write $\tilde{G}(X,0,M) = \lbrace M, G(X,0,M)_1 \rbrace$, because
>
> when $a=0$, we have $\tilde{G}(X,0,M)_0= M$
>
> when $a=1$, $A=a'$ holds, we have $\tilde{G}(X,0,M)_1= G(X,0,M)_1$
>
> (3) Answer to why derivation from equation(17) to equation (18) holds:
>
> The first term in Equation (17) equals to the first term in  Equation (18):
>
> This is because, by replacing the term  $\tilde{G}(X,0,M) = \lbrace M, G(X,0,M)_1 \rbrace$ above, $\log\big( {D}(X, \tilde{G}\left(X,0,M \right))\_{0}\big)  = \log \big( {D}(X, \lbrace M, {G}\left(X,0,M \right)\_1\rbrace)\_{0} \big) $.
>
> The second term in Equation (17) equals the second term in Equation (18). We show it below.
>
> Similarly to the send step (2) , we consider $A=1$, we can get
>
> $\tilde{G}(X,1,M) = \lbrace G(X,1,M)_0, M \rbrace$, because
>
> when $a=0$, $A=a'$ holds,  we have $\tilde{G}(X,1,M)_0= G(X,1,M)_0$
>
> when $a=1$, we have $\tilde{G}(X,0,M)_1= M$
>
> Now we come back to Equation (5) in our paper.
> Because The discriminator ${D}$ then determines which component of $\tilde{G}$ is the observed factual mediator and thus outputs the corresponding probability. Thus for the input $(X,\tilde{G}) $, the output of the discriminator $D$ is
>
> ${D}(X, \tilde{G})_a = \hat{\mathbb{P}}( M = \tilde{G}_a \mid X,  \tilde{G}) = \hat{\mathbb{P}}( A = a\mid X,  \tilde{G}) $. Because it is the corresponding probability, therefore, the sum of the ${D}(X, \tilde{G})_0$ and ${D}(X, \tilde{G})_1$ should be 1.
>
> By replacing the term $\tilde{G}(X,1,M) = \lbrace G(X,1,M)_0, M \rbrace$ which we have shown above, we can have the second term in Equation (17) equals the second term in Equation (18):
>
> $ \log \big( {D}(X, \tilde{G}\left(X,1,M \right))\_{1}\big) = \log \big(1 - {D}(X, \{\lbrace {G}\left(X,1,M \right)\_0, M \rbrace\})\_{0} \big) $.
>
> Thus, Equation(17) equals Equation (18).
>
> **Action:** We have made the derivation more detailed in the paper (see our revised **Appendix F**).

---

> ### Author Response · Authors · 2024-12-02
> **Response to Reviewer d2d9**
>
> Dear Reviewer,
>
> Thank you for your thoughtful feedback and for taking the time to review our submission. We have carefully addressed all the questions and concerns you raised in your review. We have also incorporated all action points into our revised paper.
>
> If you find our clarifications satisfactory, we would greatly appreciate it if you could consider raising your score accordingly.
>
> Thank you again for your constructive comments, which have helped improve the quality of our submission.

---

### Author Response · Authors · 2024-11-20
**Response to all reviewers**

Thank you very much for the constructive evaluation of our paper and your helpful comments! We addressed all of them in the comments below.


Our **main improvements** are the following:


* **Main contribution.** We clarified the theoretical contribution of our paper. Ours is the first work in the counterfactual fairness literature to **rigorously state the assumptions needed for the identifiability of counterfactuals in our theory and then derive identifiability results**. For this, we propose a tailored model specific to this task and further provide **theoretical guarantees** that we learn the **correct counterfactual distribution** and therefore the counterfactual fair predictions. Ours is the **first** neural method for counterfactual fairness that offers such theoretical guarantees.


* **Additional insights into how our method works.** We provide further insights into how our method operates. We explain why we can still learn the correct counterfactuals instead of reproducing factual mediators. The reason is due to the adversarial training process of our generator. We empirically prove that by experiments and additionally give three important arguments for the reason why it works (see our **new Section 5.2.2 and Appendix K.2** in our revised paper)

* **Computational efficiency.** We discuss the efficiency of our method. Our proposed framework used multiple GANs primarily to meet the mathematical assumptions and thus ensure **theoretical guarantees**. However, having multiple GANS is not necessary in practice. For this, we provide **new experimental results** to demonstrate that a **single GAN is sufficient for state-of-the-art performance**. This thus demonstrates the scalability of our method (see our **new Section 5.2.2 and Appendix K.3**  in the revised paper).


* **Assumptions for identifiability of counterfactuals.** We discuss the plausibility of the assumption for identifiability of counterfactuals in greater depth. Related works for counterfactual fairness **all act as heuristics that may return incorrect estimates**.  We explain why the lack of identifiability can lead to unfair predictions and why existing baselines can not ensure real counterfactual fairness prediction (see **Section 2.2 and Appendix C and D** ).  We thus believe that having some assumptions in our case is necessary, and is a **significant step forward toward a safe and reliable use of counterfactual fairness in practice.**

We highlighted all key changes in our revised paper in blue color. We will incorporate all changes (marked with Action) into the camera-ready version of our paper. Given these improvements, we are confident that our paper provides valuable contributions to the causal machine learning literature and is a good fit for ICLR 2025.

---

### Meta-Review · Area_Chair_6nRj · 2024-12-18

**Metareview:**

The contribution correctly points out that many papers don't acknowledge the problem of identifiability of counterfactuals, or at least assume they can be solved somehow by an existing but unspecified principle. Despite the interesting take, and the wide coverage of the area, there were still some important problems. For instance, in the answer to Q2 of d2d9, it seems $M$ must be a continuous scalar?

Concerning conceptual novelty: I found the claims that "Thereby, we can – for the first time – show in which scenarios counterfactual fairness can be fulfilled (and in which scenarios not)" (lines 876-879) to be unsubstantiated. In one of the rebuttal paragraphs, it is stated that "Additionally, paper [1] provided only the conceptual algorithm, while only later works proceeded by offering actual instantiation. Therefore, [1] did not provide any theoretical guarantees for achieving counterfactual fairness in the prediction models."

This is factually incorrect. As discussed in the original Kusner et al. paper (and the Russell et al. paper in the same 2017 NeurIPS, "When Worlds Collide"), one must rely on either identifiable latent variables not in a path from $A$ or, that failing, identifiable counterfactuals. This is a point repeated multiple times in the Russell et al. paper, and in related papers of partial identification such as Wu et al.s PC-Fairness in NeurIPS 2019. Kusner et al. explicitly use the additive error model structural equation as an example, which is a special case of BGM. It is unclear what is meant by "Kusner et al. (2017) did not clarify how to learn latent variables in practice and did not prove the identifiability of the inferred latent variables by a model." (lines 926-928). But "learning latent variables" and "prove the(ir) identifiability" are standard problems, and there was no need to reinvent the wheel. It's uncharitable to assume that an informed reader can't understand that. References were given in Kusner et al., instead of being discussed, as the literature on that is large: they cite for instance, Bollen's SEM book from 1989, in which scalar latent variables with three conditionally independent children in a linear model (and some discrete cases, see Kruskal's classical work) are known to be identifiable, the same structure found in the Law School example.

As a matter of fact, as discussed by the reviewers, the presentation of the paper could learn from this and be improved by clarifying that it does not (and need not) provide a substantive new solution to the identification problem, it is using a off-the-shelf identification principle (BGM) that happens to be different from the additive error principle used as an example in Kusner et al. It could use others. The present manuscript could modularize the idea of (1) identifying counterfactuals, and (2) presenting a learning algorithm. As is, a reader might get confused about the fact that these two ideas need not to be fundamentally connected.

Concerning the algorithm: like in the discussion among reviewers, I see the GAN method as useful, but perhaps unnecessarily intertwined with BGM, and a tad overcomplicated. For instance, within BGM-like conditions, Theorem 1 of Xie et al. "Advancing Counterfactual Inference through Nonlinear Quantile Regression" (https://arxiv.org/abs/2306.0575, ICML workshop “Could it have been different?” Counterfactuals in Minds and Machines") says we can derive all counterfactuals deterministically from observation and structure such as the one in Figure 1.  So we can train a fair classifier using $(X, M_a, M_{a'})$, with the missing counterfactuals imputed. Here, there is no direct use of $A$ nor $U$, both potential outcomes are used by the classifier without being told which one is factual (as also mentioned in Section 4 of Silva (2024), we can use the full counterfactual process $(M_a, M_{a'})$ instead of error terms in a counterfactually fair predictor).

Notice that the above is exactly the same pipeline as in the main example in Kusner et al. (2017), substituting the additive error model with BGM (much can be said about the inadequacy of BGM anyway, but I won't go into that for this assessment)

I'm sure this can be further generalized and even simplified, but a question I'd pose for a future version is why not take this route at least as an alternative method of comparison.

Another suggestion: in the rebuttal, it is claimed that

*...it can be intuitively re-formulated ([4]) as follows: In , the sensitive attribute, , is assumed to interact only with the observed covariates, , and not with the exogenous noise, . Many real-world data-generation mechanisms / phenomena, if studied closely, can be said to satisfy this assumption, e.g., simulators in physics and medicine but also neuroscience and behavioral processes.*

Having an explicit elaboration on that would be a useful (and, imho, non-trivial) insight in order to better motivate the BGM, but I don't see it.

**Additional Comments On Reviewer Discussion:**

The discussions below touch upon two main points, elaborated in more detail in the meta-review: i) the extent that the conceptual points are really novel, and the BGM assumption worth of further exploration; ii) the extent by which introducing the complex machinery of GANs does pay off.

---

### Decision · Program_Chairs · 2025-01-22

Reject